# BYZANTINE-ROBUST DECENTRALIZED LEARNING VIA CLIPPEDGOSSIP

## ABSTRACT

In this paper, we study the challenging task of Byzantine-robust decentralized training on arbitrary communication graphs. Unlike federated learning where workers communicate through a server, workers in the decentralized environment can only talk to their neighbors, making it harder to reach consensus and benefit from collaborative training. To address these issues, we propose a CLIPPEDGOSSIP algorithm for Byzantine-robust consensus and optimization, which is the first to provably converge to a $\mathcal{O}(\delta_{\max}\zeta^2/\gamma^2)$ neighborhood of the stationary point for non-convex objectives under standard assumptions. Finally, we demonstrate the encouraging empirical performance of CLIPPEDGOSSIP under a large number of attacks.

## 1 INTRODUCTION

*"Divide et impera"*.

Distributed training arises as an important topic due to privacy constraints of decentralized data storage (McMahan et al., 2017; Kairouz et al., 2019). As the server-worker paradigm suffers from a single point of failure, there is a growing amount of works on training in the absence of server (Lian et al., 2017; Nedic, 2020; Koloskova et al., 2020b). We are particularly interested in decentralized scenarios where direct communication may be unavailable due to physical constraints. For example, devices in a sensor network can only communicate devices within short physical distances.

Failures—from malfunctioning or even malicious participants—are ubiquitous in all kinds of distributed computing. A *Byzantine* adversarial worker can deviate from the prescribed algorithm and send arbitrary messages and is assumed to have the knowledge of the whole system (Lamport et al., 2019). It means Byzantine workers not only collude, but also know the data, algorithm, and models of all regular workers. However, they cannot directly modify the states on regular workers, nor compromise messages sent between two connected regular workers.

Defending Byzantine attacks in a communication-constrained graph is challenging. As secure broadcast protocols are no longer available (Pease et al., 1980; Dolev & Strong, 1983; Hirt & Raykov, 2014), regular workers can only utilize information from their own neighbors who have heterogeneous data distribution or are malicious, making it very difficult to reach global consensus. While there are some works attempt to solve this problem (Su & Vaidya, 2016a; Sundaram & Gharesifard, 2018), their strategies suffer from serious drawbacks: 1) they require regular workers to be very densely connected; 2) they only show asymptotic convergence or no convergence proof; 3) there is no evidence if their algorithms are better than training alone.

In this work, we study the Byzantine-robustness decentralized training in a constrained topology and address the aforementioned issues. The main contributions of our paper are summarized as follows:

- We identify a novel network robustness criterion, characterized in terms of the spectral gap of the topology ($\gamma$) and the number of attackers ($\delta$), for consensus and decentralized training, applying to a much broader spectrum of graphs than (Su & Vaidya, 2016a; Sundaram & Gharesifard, 2018).
- We propose CLIPPEDGOSSIP as the defense strategy and provide, for the first time, precise rates of robust convergence to a $\mathcal{O}(\delta_{\max}\zeta^2/\gamma^2)$ neighborhood of a stationary point for stochastic objectives under standard assumptions.[1] We also empirically demonstrate the advantages of CLIPPEDGOSSIP over previous works.
- Along the way, we also obtain the fastest convergence rates for standard non-robust (Byzantine-free) decentralized stochastic non-convex optimization by using local worker momentum.

---

[1] In a previous version, we referred to CLIPPEDGOSSIP as *self-centered clipping*.

## 2 RELATED WORK

Recently there have been extensive works on Byzantine-resilient distributed learning with a trustworthy server. The statistics-based robust aggregation methods cover a wide spectrum of works including median (Chen et al., 2017; Blanchard et al., 2017; Yin et al., 2018; Mhamdi et al., 2018; Xie et al., 2018; Yin et al., 2019), geometric median (Pillutla et al., 2019), signSGD (Bernstein et al., 2019; Li et al., 2019; yong Sohn et al., 2020), clipping (Karimireddy et al., 2021a;b), and concentration filtering (Alistarh et al., 2018; Allen-Zhu et al., 2020; Data & Diggavi, 2021). Other works explore special settings where the server owns the entire training dataset (Xie et al., 2020a; Regatti et al., 2020; Su & Vaidya, 2016b; Chen et al., 2018; Rajput et al., 2019; Gupta et al., 2021). The state-of-the-art attacks take advantage of the variance of good gradients and accumulate bias over time (Baruch et al., 2019; Xie et al., 2019). A few strategies have been proposed to provably defend against such attacks, including momentum (Karimireddy et al., 2021a; El Mhamdi et al., 2021) and concentration filtering (Allen-Zhu et al., 2021).

Decentralized machine learning has been extensively studied in the past few years (Lian et al., 2017; Koloskova et al., 2020b; Li et al., 2021; Ying et al., 2021b; Lin et al., 2021; Kong et al., 2021; Yuan et al., 2021; Kovalev et al., 2021). The state-of-the-art convergence rate is established in (Koloskova et al., 2020b) is $\mathcal{O}(\frac{\sigma^2}{n\varepsilon^2} + \frac{\sigma}{\sqrt{\gamma}\varepsilon^{3/2}})$ where the leading $\frac{\sigma^2}{n\varepsilon^2}$ is optimal. In this paper we improve this rate to $\mathcal{O}(\frac{\sigma^2}{n\varepsilon^2} + \frac{\sigma^{2/3}}{\gamma^{2/3}\varepsilon^{4/3}})$ using local momentum.

Decentralized machine learning with certified Byzantine-robustness is less studied. When the communication is unconstrained, there exist secure broadcast protocols that guarantee all regular workers have identical copies of each other's update (Gorbunov et al., 2021; El-Mhamdi et al., 2021). We are interested in a more challenging scenario where not all workers have direct communication links. In this case, regular workers may behave very differently depending on their neighbors in the topology. One line of work constructs a Public-Key Infrastructure (PKI) so that the message from each worker can be authenticated using digital signatures. However, this is very inefficient requiring quadratic communication (Abraham et al., 2020). Further, it also requires every worker to have a globally unique identifier which is known to every other worker. This assumption is rendered impossible on general communication graphs, motivating our work to explicitly address the graph topology in decentralized training. Sybil attacks are an important orthogonal issue where a single Byzantine node can create innumerable "fake nodes" overwhelming the network (cf. recent overview by Ford (2021)). Truly decentralized solutions to this are challenging and sometimes rely on heavy machinery, e.g. blockchains (Poupko et al., 2021) or Proof-of-Personhood (Borge et al., 2017).

More related to the approaches we study, Su & Vaidya (2016a); Sundaram & Gharesifard (2018); Yang & Bajwa (2019b;a) use trimmed mean at each worker to aggregate models of its neighbors. This approach only works when all regular workers have an honest majority among their neighbors and are densely connected. Guo et al. (2021) evaluate the incoming models of a good worker with its local samples and only keep those well-perform models for its local update step. However, this method only works for IID data. Peng & Ling (2020) reformulate the original problem by adding TV-regularization and propose a GossipSGD type algorithm which works for strongly convex and non-IID objectives. However, its convergence guarantees are inferior to non-parallel SGD. In this work, we address all of the above issues and are able to provably relate the communication graph (spectral gap) with the fraction of Byzantine workers. Besides, most works do not consider attacks that exploit communication topology, except (Peng & Ling, 2020) who propose zero-sum attack. We defer detailed comparisons and more related works to § F.

## 3 SETUP

### 3.1 DECENTRALIZED THREAT MODEL

Consider an undirected graph $\mathcal{G} = (\mathcal{V}, \mathcal{E})$ where $\mathcal{V} = \{1, \ldots, n\}$ denotes the set of workers and $\mathcal{E}$ denotes the set of edges. Let $\mathcal{N}_i \subset \mathcal{V}$ be the neighbors of node $i$ and $\overline{\mathcal{N}}_i := \mathcal{N}_i \cup \{i\}$. In addition, we assume there are no self-loops and the system is synchronous. Let $\mathcal{V}_B \subset \mathcal{V}$ be the set of Byzantine workers with $b = |\mathcal{V}_B|$ and the set of regular (non-Byzantine) workers is $\mathcal{V}_R := \mathcal{V}\backslash\mathcal{V}_B$. Let $\mathcal{G}_R$ be the subgraph of $\mathcal{G}$ induced by the regular nodes $\mathcal{V}_R$ which means removing all Byzantine nodes and their associated edges. If the reduced graph $\mathcal{G}_R$ is disconnected, then there exist two regular workers who cannot reliably exchange information. In this setting, training on the combined data of all the good workers is impossible. Hence, we make the following necessary assumption.

**(A1) Connectivity.** *$\mathcal{G}_R$ is connected.*

**Remark 1.** *In contrast, Su & Vaidya (2016a); Sundaram & Gharesifard (2018) impose a much stronger assumption that the subgraph of $\mathcal{G}_R$ of the regular workers remain connected even after additionally removing any $|\mathcal{V}_B|$ number of edges. For example, the graph in Fig. 1 with 1 Byzantine worker $V_1$ satisfies (A1) but does not satisfy their assumption as removing an additional edge at $A_1$ or $B_1$ may discard the graph cut.*

In decentralized learning, each regular worker $i \in \mathcal{V}_R$ locally stores a vector $\{\boldsymbol{W}_{ij}\}_{j=1}^n$ of mixing weights, for how to aggregate model updates received from neighbors. We make the following assumption on the weight vectors.

**(A2) Mixing weights.** *The weight vectors on regular workers satisfy the following properties:*

- *Each regular worker $i \in \mathcal{V}_R$ stores non-negative $\{\boldsymbol{W}_{ij}\}_{j=1}^n$ with $\boldsymbol{W}_{ij} > 0$ iff $j \in \overline{\mathcal{N}}_i$;*
- *The adjacent weights to each regular worker $i \in \mathcal{V}_R$ sum up to 1, i.e. $\sum_{j=1}^n \boldsymbol{W}_{ij} = 1$;*
- *For $i, j \in \mathcal{V}_R$, $\boldsymbol{W}_{ij} = \boldsymbol{W}_{ji}$.*

We can construct such weights even in the presence of Byzantine workers, using algorithms that only rely on communication with local neighbors, e.g. Metropolis-Hastings (Hastings, 1970). We defer details of the construction to § C.2. Note that the Byzantine workers $\mathcal{V}_B$ might also obtain such weights, however, they can use arbitrary different weights in reality during the training.

We define $\delta_i := \sum_{j \in \mathcal{V}_B} \boldsymbol{W}_{ij}$ to be the total weight of adjacent Byzantine edges around a regular worker $i$, and define the maximum Byzantine weight as $\delta_{\max} := \max_{i \in \mathcal{V}_R} \delta_i$.

**Remark 2.** *In the decentralized setting, the total fraction of Byzantine nodes $|\mathcal{V}_B|/n$ is irrelevant. Instead, what matters is the fraction of the edge weights they control which are adjacent to regular nodes (as defined by $\delta_i$ and $\delta_{\max}$). This is because a Byzantine worker can send different messages along each edge. Thus, a single Byzantine worker connected to all other workers with large edge weights can have a large influence on all the other workers. Similarly, a potentially very large number of Byzantine workers may overall have very little effect—if the edges they control towards good nodes have little weight. When we have a uniform fully connected graph (such as in the centralized setting), the two notions of bad nodes & edges become equivalent.*

To facilitate our analysis of convergence rate, we define a *hypothetical* mixing matrix $\widetilde{\boldsymbol{W}} \in \mathbb{R}^{(n-b)\times(n-b)}$ for the subgraph $\mathcal{G}_\mathcal{R}$ of regular workers with entry $i, j \in \mathcal{V}_R$ defined as

$$\widetilde{\boldsymbol{W}}_{ij} = \begin{cases} \boldsymbol{W}_{ij} & \text{if } i \neq j \\ \boldsymbol{W}_{ii} + \delta_i & \text{if } i = j. \end{cases} \tag{1}$$

By the construction of this hypothetical matrix $\widetilde{\boldsymbol{W}}$, the following property directly follows.

**Lemma 3.** *Given (A2), then $\widetilde{\boldsymbol{W}}$ is symmetric and doubly stochastic, i.e.*

$$\widetilde{\boldsymbol{W}}_{ij} = \widetilde{\boldsymbol{W}}_{ji}, \ \sum_{i=1}^n \widetilde{\boldsymbol{W}}_{ij} = 1, \ \sum_{j=1}^n \widetilde{\boldsymbol{W}}_{ij} = 1. \qquad \forall i, j \in [n-b]$$

Further, the spectral gap of the matrix $\widetilde{\boldsymbol{W}}$ is positive.

**Lemma 4.** *By (A1) and (A2), there exists $\gamma \in (0, 1]$ such that $\forall \ \boldsymbol{x} \in \mathbb{R}^{n-b}$ and $\bar{\boldsymbol{x}} = \frac{\mathbf{1}^\top \boldsymbol{x}}{n-b}\mathbf{1} \in \mathbb{R}^{n-b}$*

$$\|\widetilde{\boldsymbol{W}}\boldsymbol{x} - \bar{\boldsymbol{x}}\|_2 \leq (1-\gamma)\|\boldsymbol{x} - \bar{\boldsymbol{x}}\|_2. \tag{2}$$

The $\gamma(\widetilde{\boldsymbol{W}})$ is the *spectral gap* of the subgraph of regular workers $\mathcal{G}_R$. We have $\gamma = 0$ if and only if $\mathcal{G}_R$ is disconnected, and $\gamma = 1$ if and only if $\mathcal{G}_R$ is fully connected.

In summary, $\gamma$ measures the connectivity of the regular subgraph $\mathcal{G}_R$ formed after removing the Byzantine nodes, whereas $\delta_i$ and $\delta_{\max}$ are a measure of the influence of the Byzantine nodes.

## 3.2 OPTIMIZATION ASSUMPTIONS

We study the general distributed optimization problem

$$\min_{\boldsymbol{x} \in \mathbb{R}^d} f(\boldsymbol{x}) := \frac{1}{|\mathcal{V}_R|} \sum_{i \in \mathcal{V}_R} \left\{ f_i(\boldsymbol{x}) := \mathbb{E}_{\xi_i \sim \mathcal{D}_i} F_i(\boldsymbol{x}; \xi_i) \right\} \tag{3}$$

on heterogeneous (non-IID) data, where $f_i$ is the local objective on worker $i$ with data distribution $\mathcal{D}_i$ and independent noise $\xi_i$. We assume that the gradients computed over these data distributions satisfy the following standard properties.

**(A3) Bounded noise and heterogeneity.** *Assume that for all $i \in \mathcal{V}_R$ and $\boldsymbol{x} \in \mathbb{R}^d$, we have*

$$\mathbb{E}_{\xi \sim \mathcal{D}_i} \|\nabla F_i(\boldsymbol{x}; \xi) - \nabla f_i(\boldsymbol{x})\|^2 \leq \sigma^2, \qquad \mathbb{E}_{j \sim \mathcal{V}_R} \|\nabla f_j(\boldsymbol{x}) - \nabla f(\boldsymbol{x})\|^2 \leq \zeta^2. \qquad (4)$$

**(A4) L-smoothness.** *For $i \in \mathcal{V}_R$, $f_i(\boldsymbol{x}) : \mathbb{R}^d \to \mathbb{R}$ is differentiable and there exists a constant $sL \geq 0$ such that for each $\boldsymbol{x}, \boldsymbol{y} \in \mathbb{R}^d$:*

$$\|\nabla f_i(\boldsymbol{x}) - \nabla f_i(\boldsymbol{y})\| \leq L\|\boldsymbol{x} - \boldsymbol{y}\|. \qquad (5)$$

We denote $\boldsymbol{x}_i^t \in \mathbb{R}^d$ as the state of worker $i \in \mathcal{V}_R$ at time $t$.

# 4 ROBUST DECENTRALIZED CONSENSUS

Agreeing on one value (*consensus*) among regular workers is one of the fundamental questions in distributed computing. *Gossip averaging* is a common consensus algorithm in the Byzantine-free case ($\delta = 0$). Applying gossip averaging steps iteratively to all nodes formally writes as

$$\boldsymbol{x}_i^{t+1} := \sum_{j=1}^n \boldsymbol{W}_{ij} \boldsymbol{x}_j^t, \qquad t = 0, 1, \ldots \qquad \text{(GOSSIP)}$$

Suppose each worker $i \in [n]$ initially owns a different $\boldsymbol{x}_i^0$ and (A1) and (A2) hold true, then each worker's iterate $\boldsymbol{x}_i^t$ asymptotically converges to $\boldsymbol{x}_i^\infty = \bar{\boldsymbol{x}} = \frac{1}{n} \sum_{j=1}^n \boldsymbol{x}_j^0$, for all $i \in [n]$, which is also known as average consensus (Boyd et al., 2006). Reaching consensus in the presence of Byzantine workers is more challenging, with a long history of study (LeBlanc et al., 2013; Su & Vaidya, 2016a).

## 4.1 THE CLIPPED GOSSIP ALGORITHM

We introduce a novel decentralized gossip-based aggregator, termed CLIPPEDGOSSIP, for Byzantine-robust consensus. CLIPPEDGOSSIP uses its local reference model as center and clips all received neighbor model weights. Formally, for $\text{CLIP}(\boldsymbol{z}, \tau) := \min(1, \tau/\|\boldsymbol{z}\|) \cdot \boldsymbol{z}$, we define for node $i$

$$\boldsymbol{x}_i^{t+1} := \sum_{j=1}^n \boldsymbol{W}_{ij}(\boldsymbol{x}_i^t + \text{CLIP}(\boldsymbol{x}_j^t - \boldsymbol{x}_i^t, \tau_i)), \qquad t = 0, 1, \ldots \qquad \text{(CLIPPEDGOSSIP)}$$

**Theorem I.** *Let $\bar{\boldsymbol{x}}^t := \frac{1}{|\mathcal{V}_R|} \sum_{i \in \mathcal{V}_R} \boldsymbol{x}_i^t$ be the average iterate over the unknown set of regular nodes. If the initial consensus distance is bounded as $\frac{1}{|\mathcal{V}_R|} \sum_{i \in \mathcal{V}_R} \mathbb{E}\|\boldsymbol{x}_i^t - \bar{\boldsymbol{x}}^t\|^2 \leq \rho^2$, then for all $i \in \mathcal{V}_R$, the output $\boldsymbol{x}_i^{t+1}$ of CLIPPEDGOSSIP with an appropriate choice of clipping radius satisfies*

$$\frac{1}{|\mathcal{V}_R|} \sum_{i \in \mathcal{V}_R} \mathbb{E}\|\boldsymbol{x}_i^{t+1} - \bar{\boldsymbol{x}}^t\|^2 \leq \left(1 - \gamma + c\sqrt{\delta_{\max}}\right)^2 \rho^2 \qquad \text{and} \quad \mathbb{E}\|\bar{\boldsymbol{x}}^{t+1} - \bar{\boldsymbol{x}}^t\|^2 \leq c^2 \delta_{\max} \rho^2$$

*where the expectation is over the random variable $\{\boldsymbol{x}_i^t\}_{i \in \mathcal{V}_R}$ and $c > 0$ is a constant.*

We inspect Theorem I on corner cases. If regular workers have already reached consensus before aggregation ($\rho = 0$), then Theorem I shows that we retain consensus even in the face of Byzantine agents. In this case, we can use a simple majority, which corresponds to setting clipping threshold $\tau_i = 0$. Further, if there is no Byzantine worker ($\delta_{\max} = 0$), then the robust aggregator must improve the consensus distance by a factor of $(1 - \gamma)^2$ which matches standard gossiping analysis (Boyd et al., 2006). Finally, for the complete graph ($\gamma = 1$) CLIPPEDGOSSIP satisfies the centralized notion of ($\delta_{\max}, c^2$)-robust aggregator in (Karimireddy et al., 2021a, Definition C). Thus, CLIPPEDGOSSIP recovers all past optimal aggregation methods as special cases.

Note that if the topology is poorly connected and there are Byzantine attackers with ($\gamma < c\sqrt{\delta_{\max}}$), then Theorem I gives no guarantee that the consensus distance will reduce after aggregation. This is unfortunately not possible to improve upon, as we will show in the following § 4.2—if the connectivity is poor then the effect of Byzantine workers can be significantly amplified.

## 4.2 LOWER BOUNDS DUE TO COMMUNICATION CONSTRAINTS

Not all pairs of workers have direct communication links due to constraints such as physical distances in a sensor network. It is common that a subset of sensors are clustered within a small physical space while only few of them have communication links to the rest of the sensors. Such links form a cut-set of the communication topology and are crucial for information diffusion. On the other hand, attackers can increase consensus errors in the presence of these critical links.

**Theorem II.** *Consider networks satisfying (A1) of $n$ nodes, each holding a number in $\{0, 1\}$, and only $\mathcal{O}(1/n^2)$ of the edges are adjacent to attackers. For any robust consensus algorithm $\mathcal{A}$, there exists a network such that the output of $\mathcal{A}$ has an average consensus error of at least $\Omega(1)$.*

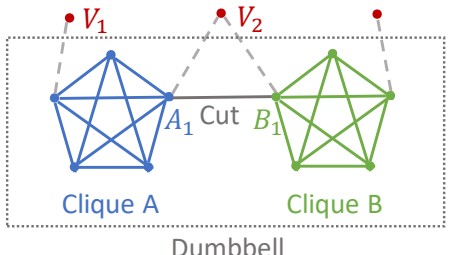

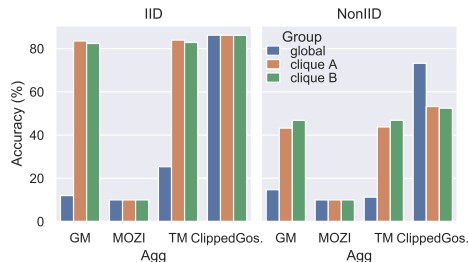

Figure 1: A dumbbell topology of two cliques A and B of regular workers connected by an edge (graph cut). Byzantine workers (red) may attack the graph at different places.

Figure 2: Accuracies of models trained with robust aggregators over dumbbell topology and CIFAR-10 dataset ($\delta = 0$). The models are averaged within clique A, B, or all regular workers separately.

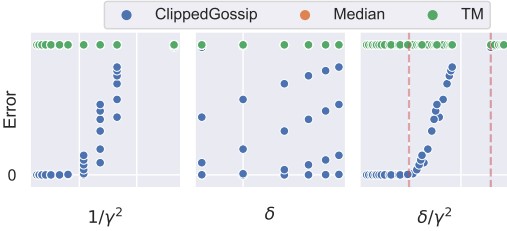

Figure 3: Performance of CLIPPEDGOSSIP and baselines (TM and MEDIAN) under Byzantine attacks with varying $\gamma$ and $\delta_{\max}$. Each point represents the squared average consensus error of the last iterate of an algorithm. MEDIAN and TM have identical performance and CLIPPEDGOSSIP is consistently better. Further, the performance of CLIPPEDGOSSIP is best explained by the magnitude of $(\delta/\gamma^2)$ – it is excellent when the ratio is less than a threshold and degrades as it increases.

*Proof.* Consider two cliques A and B with $n$ nodes each connected by an edge to each other and to a Byzantine node $V_2$, c.f. Fig. 1. Suppose that we know all nodes have values in $\{0, 1\}$. Let all nodes in A have value 0. Now consider two settings:

**World 1.** All B nodes have value 0. However, Byzantine node $V_2$ pretends to be part of a clique identical to B which it *simulates*, except that all nodes have value 1. The true consensus average is 0.
**World 2.** All B nodes have value 1. This time the Byzantine node $V_2$ simulates clique B with value 0. The true consensus average here is 0.5.

From the perspective of clique A, the two worlds are identical–it seems to be connected to one clique with value 0 and another with value 1. Thus, it must make $\Omega(1)$ error at least in one of the worlds. This proves that consensus is impossible in this setting. □

While arguments above are similar to classical lower bounds in decentralized consensus which show we need $\delta \leq 1/3$ (Fischer et al., 1986), in our case there is only 1 Byzantine node (out of $2n + 1$ regular nodes) which controls only 2 edges i.e. $\delta = \mathcal{O}(1/n^2)$. This impossibility result thus drives home the additional impact through the restricted communication topology. Further, past impossibility results about robust decentralized consensus such as (Sundaram & Gharesifard, 2018; Su & Vaidya, 2016a) use combinatorial concepts such as the number of node-disjoint paths between the good nodes. However, such notions cannot account for the edge weights easily and cannot give finite-time convergence guarantees. Instead, our theory shows that the ratio of $\delta_{\max}/\gamma^2$ accurately captures the difficulty of the problem. We next verify this empirically.

In Fig. 3, we show the final consensus error of three defenses under Byzantine attacks. TM and MEDIAN have a large error even for small $\delta_{\max}$ and large $\gamma$. The consensus error of CLIPPEDGOSSIP increases almost linearly with $\delta_{\max}/\gamma^2$. However, this phenomenon is not observed by looking at $\gamma^{-2}$ or $\delta_{\max}$ alone, validating our theoretical analysis in Theorem I. Details are deferred to § D.1.

---

**Algorithm 1** Byzantine-Resilient Decentralized Optimization with CLIPPEDGOSSIP

---

**Input:** $\boldsymbol{x}^0 \in \mathbb{R}^d$, $\alpha$, $\eta$, $\{\tau_i^t\}$, $\boldsymbol{m}_i^0 = \boldsymbol{g}_i(\boldsymbol{x}^0)$
1: **for** $t = 0, 1, \ldots$ **do**
2:     **for** $i = 1, \ldots, n$ **in parallel**
3:         $\boldsymbol{m}_i^{t+1} = (1 - \alpha)\boldsymbol{m}_i^t + \alpha \boldsymbol{g}_i(\boldsymbol{x}_i^t)$
4:         $\boldsymbol{x}_i^{t+1/2} = \boldsymbol{x}_i^t - \eta \boldsymbol{m}_i^{t+1}$ if $i \in \mathcal{V}_{\mathsf{R}}$ else $*$
5:         Exchange $\boldsymbol{x}_i^{t+1/2}$ with $\mathcal{N}_i$
6:         $\boldsymbol{x}_i^{t+1} = \text{CLIPPEDGOSSIP}_i(\boldsymbol{x}_1^{t+1/2}, \ldots, \boldsymbol{x}_n^{t+1/2}; \tau_i^{t+1})$
7:     **end for**

---

Table 1: Comparison with prior work of convergence rates for non-convex objectives to a $\mathcal{O}(\delta\zeta^2)$-neighborhood of stationary points. We recover comparable or improved rates as special cases.

| | Reference | Setting | Convergence to $\varepsilon$-accuracy |
|---|---|---|---|
| Regular ($\delta = 0$) Decentralized | Koloskova et al. (2020b) | - | $\mathcal{O}\left(\frac{\sigma^2}{n\varepsilon^2} + \frac{\zeta}{\gamma\varepsilon^{3/2}} + \frac{\sigma}{\sqrt{\gamma}\varepsilon^{3/2}} + \frac{1}{\gamma\varepsilon}\right)$ |
| | This work | $\delta = 0$ | $\mathcal{O}\left(\frac{\sigma^2}{n\varepsilon^2} + \frac{\zeta}{\gamma\varepsilon^{3/2}} + \frac{\sigma^{2/3}}{\gamma^{2/3}\varepsilon^{4/3}} + \frac{1}{\gamma\varepsilon}\right)$ |
| Byzantine-robust Fully-connected ($\gamma = 1$) IID ($\zeta = 0$) | Guo et al. (2021) | - | ✗ |
| | Gorbunov et al. (2021) | $\delta$ known | $\mathcal{O}\left(\frac{\sigma^2}{n\varepsilon^2} + \frac{n\delta\sigma^2}{m\varepsilon} + \frac{1}{\varepsilon}\right)$ † |
| | Gorbunov et al. (2021) | $\delta$ unknown | $\mathcal{O}\left(\frac{\sigma^2}{n\varepsilon^2} + \frac{n^2\delta\sigma^2}{m\varepsilon} + \frac{1}{\varepsilon}\right)$ † |
| | This work | $\gamma = 1, \zeta = 0$ | $\mathcal{O}\left(\frac{\sigma^2}{n\varepsilon^2} + \frac{\delta\sigma^2}{\varepsilon^2} + \frac{1}{\varepsilon}\right)$ |
| Byzantine-robust Federated Learning | Karimireddy et al. (2021b) | - | $\mathcal{O}\left(\frac{\sigma^2}{\varepsilon^2}(\delta + \frac{1}{n}) + \frac{1}{\varepsilon}\right)$ |
| | This work | $\gamma = 1$ | $\mathcal{O}\left(\frac{\sigma^2}{\varepsilon^2}(\delta + \frac{1}{n}) + \frac{\zeta}{\varepsilon^{3/2}} + \frac{\sigma^{2/3}}{\varepsilon^{4/3}} + \frac{1}{\varepsilon}\right)$ |

† This method does not generalize to constrained communication topologies.

## 5 ROBUST DECENTRALIZED OPTIMIZATION

The general decentralized training algorithm can be formulated as

$$\boldsymbol{x}_i^{t+1/2} := \begin{cases} \boldsymbol{x}_i^t - \eta \boldsymbol{g}_i(\boldsymbol{x}_i^t) & i \in \mathcal{V}_{\mathsf{R}} \\ * & i \in \mathcal{V}_{\mathsf{B}} \end{cases}, \qquad \boldsymbol{x}_i^{t+1} := \text{AGG}_i(\{\boldsymbol{x}_k^{t+1/2} : k \in \overline{\mathcal{N}}_i\})$$

where $\eta$ is the learning rate, $\boldsymbol{g}_i(\boldsymbol{x}) := \nabla F(\boldsymbol{x}, \xi_i)$ is a stochastic gradient, and $\xi_i^t \sim \mathcal{D}_i$ is the random batch at time $t$ on worker $i$. The received message $\boldsymbol{x}_k^{t+1/2}$ can be arbitrary for Byzantine nodes $k \in \mathcal{V}_{\mathsf{B}}$. Replacing AGG with plain gossip averaging (GOSSIP) recovers standard gossip SGD (Koloskova et al., 2019). Under the presence of Byzantine workers, which is the main interest of our work, we will show that we can replace AGG with CLIPPEDGOSSIP and use local worker momentum to achieve Byzantine robustness (Karimireddy et al., 2021a). The full procedure is described in Algorithm 1.

**Theorem III.** *Suppose Assumptions 1–4 hold and $\delta_{\max} = \mathcal{O}(\gamma^2)$. Then for $\alpha := 3\eta L$, Algorithm 1 reaches $\frac{1}{T+1}\sum_{t=0}^{T}\|\nabla f(\bar{\boldsymbol{x}}^t)\|_2^2 \leq \frac{\delta_{\max}\zeta^2}{\gamma^2} + \varepsilon$ in iteration complexity*

$$\mathcal{O}\left(\frac{\sigma^2}{n\varepsilon^2}\left(\frac{1}{n} + \delta_{\max}\right) + \frac{\zeta}{\gamma\varepsilon^{3/2}} + \frac{\sigma^{2/3}}{\gamma^{2/3}\varepsilon^{4/3}} + \frac{1}{\gamma\varepsilon}\right).$$

*Furthermore, the consensus distance satisfies the upper bound*

$$\frac{1}{|\mathcal{V}_{\mathsf{R}}|}\sum_{i \in \mathcal{V}_{\mathsf{R}}}\|\boldsymbol{x}_i^T - \bar{\boldsymbol{x}}^T\|_2^2 \leq \mathcal{O}\left(\frac{\zeta^2}{\gamma^2(T+1)}\right).$$

We compare our analysis with existing works for non-convex objectives in Table 1.

**Regular decentralized training.** Even if there are no Byzantine workers ($\delta_{\max} = 0$), our convergence rate is slightly faster than that of standard gossip SGD (Koloskova et al., 2020b). The difference is that our third term $\mathcal{O}\left(\frac{\sigma^{2/3}}{\gamma^{2/3}\varepsilon^{4/3}}\right)$ is faster than their $\mathcal{O}\left(\frac{\sigma}{\sqrt{\gamma}\varepsilon^{3/2}}\right)$ for large $\sigma$ and small $\varepsilon$. This is because we use local momentum which reduces the effect of variance $\sigma$. Thus momentum has a double use in this paper in achieving robustness as well as accelerating optimization.

**Byzantine-robust federated learning.** Federated learning uses a fully connected graph ($\gamma = 1$). We compare state of the art federated learning method (Karimireddy et al., 2021b) with our rate when $\gamma = 1$. Both algorithms converge to a $\Theta(\delta\zeta^2)$-neighborhood of a stationary point and share the same leading term. This neighborhood can be circumvented with strong growth condition and over-parameterized models (Karimireddy et al., 2021b, Theorem III). We incur additional higher-order terms $\mathcal{O}(\frac{\zeta}{\gamma\varepsilon^{3/2}} + \frac{\sigma^{2/3}}{\gamma^{2/3}\varepsilon^{4/3}})$ as a penalty for the generality of our analysis. This shows that the trusted server in federated learning can be removed without significant slowdowns.

**Byzantine-robust decentralized SGD with fully connected topology**. If we limit our analysis to a special case of a fully connected graph ($\gamma = 1$) and IID data ($\zeta = 0$), then our rate has the same leading term as (Gorbunov et al., 2021), which enjoys the scaling of the total number of regular nodes. The second term $\mathcal{O}(\frac{n}{m}\frac{\delta\sigma^2}{\varepsilon})$ of (Gorbunov et al., 2021) is better than our $\mathcal{O}(\frac{1}{\varepsilon}\frac{\delta\sigma^2}{\varepsilon})$ for small $\varepsilon$ because they additionally validate $m$ random updates in each step. However, (Gorbunov et al., 2021) relies on secure protocols which do not easily generalize to constrained communication.

**Byzantine-robust decentralized SGD with constrained communication**. MOZI (Guo et al., 2021) does not provide a theoretical analysis on convergence and TM (Sundaram & Gharesifard, 2018; Su & Vaidya, 2016a; Yang & Bajwa, 2019a) only prove the asymptotic convergence of full gradient under a very strong assumption on connectivity and local honest majority.[2] Peng & Ling (2020) don't prove a rate for non-convex objective; but Gorbunov et al. (2021) which shows convergence of (Peng & Ling, 2020) on strongly convex objectives at a rate inferior to parallel SGD. In contrast, our convergence rate matches the standard stochastic analysis under much weaker assumptions than Sundaram & Gharesifard (2018); Su & Vaidya (2016a); Yang & Bajwa (2019a). Unlike these prior works, our guarantees hold even if some subsets of nodes are surrounded by a majority of Byzantine attackers. This can also be observed in practice, as we show in § D.2.3.

**Consensus for Byzantine-robust decentralized optimization.** Theorem III gives a non-trivial result that regular workers reach consensus under the CLIPPEDGOSSIP aggregator. In Fig. 2 we demonstrate the consensus behavior of robust aggregators on the CIFAR-10 dataset on a dumbbell topology, without attackers ($\delta = 0$). We compare the accuracies of models averaged within cliques A and B with model averaged over all workers. In the IID setting, the clique-averaged models of GM and TM are over 80% accuracy but the globally-averaged models are less than 30% accuracy. It means clique A and clique B are converging to two different critical points and GM and TM fail to reach consensus within the entire network! In contrast, the globally-averaged model of CLIPPEDGOSSIP is as good as or better than the clique-averaged models, both in the IID and non-IID setting.

Finally, we point out some avenues for further improvement: our results depend on the worst-case $\delta_{\max}$. We believe it is possible to replace it with a (weighted) average of the $\{\delta_i\}$ instead. Also, extending our protocols to time-varying topologies would greatly increase their practicality.

**Remark 5** (Adaptive choice of clipping radius $\tau_i^t$). *In § D.5, we give an adaptive rule to choose the clipping radius $\tau_i^t$ for all $i \in \mathcal{V}_\mathsf{R}$ and times $t$, based on the top percentile of close neighbors. This adaptive rule results in a value $\tau_i^t$ slightly smaller than the required theoretical value to preserve Byzantine robustness. In experiments, we found that the performance of optimization is robust to small perturbations of the clipping radius and that the adaptive rule performs well in all cases.*

## 6 EXPERIMENTS

In this section, we empirically demonstrate successes and failures of decentralized training in the presence of Byzantine workers, and compare the performance of CLIPPEDGOSSIP with existing robust aggregators: 1) geometric median GM (Pillutla et al., 2019); 2) coordinate-wise trimmed mean TM (Yang & Bajwa, 2019a); 3) MOZI (Guo et al., 2020). Coordinate-wise median (Yin et al., 2018) and Krum (Blanchard et al., 2017) usually perform worse than GM so we exclude them in the experiments. All implementations are based on PyTorch (Paszke et al., 2019) and evaluated on different graph topologies, with a distributed MNIST dataset (LeCun & Cortes, 2010). We defer the experiments on CIFAR10 (Krizhevsky et al., 2009) to § D.3. [3]

We defer details of robust aggregators to § A, attacks to § B, topologies and mixing matrix to § C and experiment setups and additional experiments to § D.

---

[2]MOZI is renamed to UBAR in the latest version.

[3]The code is available at this anonymous repository.

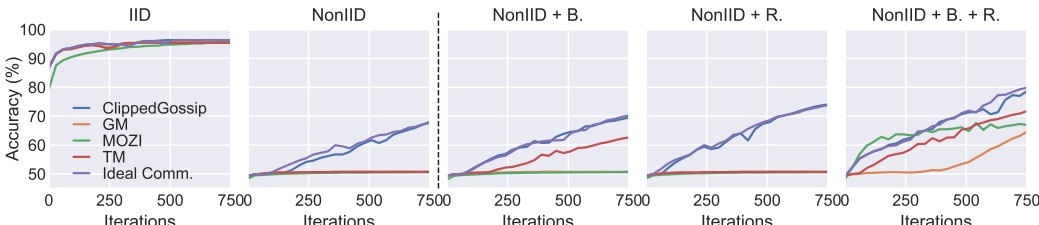

Figure 4: Accuracy of the averaged model in clique A for the dumbbell topology. In the plot title "B." stands for the bucketing (aggregating means of bucketed values) and "R." stands for adding 1 additional random edge between two cliques. We see that i) CLIPPEDGOSSIP is consistently the best matching ideal averaging performance, ii) performance mildly improves by using bucketing, and iii) significantly improves when adding a single random edge (thereby improving connectivity).

## 6.1 DECENTRALIZED DEFENSES WITHOUT ATTACKERS

Challenging topologies and data distribution may prevent existing robust aggregators from reaching consensus even when there is no Byzantine worker ($\delta = 0$). In this part, we consider the "dumbbell" topology c.f. Fig. 1. As non-IID data distribution, we split the training dataset by labels such that workers in clique A are training on digits 0 to 4 while workers in clique B are training on digits 5 to 9. This entanglement of topology and data distribution is motivated by realistic geographic constraints such as continents with dense intra-connectivity but sparse inter-connection links e.g. through an undersea cable. In Fig. 4 we compare CLIPPEDGOSSIP with existing robust aggregators GM, TM, MOZI in terms of their accuracies of averaged model in clique A. The ideal communication refers to aggregation with gossip averaging.

**Existing robust aggregators impede information diffusion.** When cliques A and B have distinct data distribution (non-IID), workers in clique A rely on the graph cut to access the full spectrum of data and attain good performance. However, existing robust aggregators in clique A completely discard information from clique B because: 1) clique B model updates are outliers to clique A due to data heterogeneity; 2) clique B updates are outnumbered by clique A updates — clique A can only observe 1 update from B due to constrained communication. The 2nd plot in Fig. 4 shows that GM, TM, and MOZI only reach 50% accuracy in the non-IID setting, supporting that they impede information diffusion. This is in contrast to the 1st plot where cliques A and B have identical data distribution (IID) and information on clique A alone is enough to attain good performance. However, reaching local models does not imply reaching consensus, c.f. Fig. 2. On the other hand, CLIPPEDGOSSIP is the only robust aggregator that preserves the information diffusion rate as the ideal gossip averaging.

**Techniques that improve information diffusion.** To address these issues, we locally employ the *bucketing* technique of (Karimireddy et al., 2021b) for the non-IID case in the 3rd subplot. Plots 4 and 5 demonstrate the impact of one additional edge between the cliques to improve the spectral gap.

- The bucketing technique randomly inputs received vectors into buckets of equal size, averages the vectors in each bucket, and finally feeds the averaged vectors to the aggregator. While bucketing helps TM to overcome 50% accuracy, TM is still behind CLIPPEDGOSSIP. GM only improves by 1% while MOZI remains at almost the same accuracy.
- Adding one more random edge between two cliques improves the spectral gap $\gamma$ from 0.0154 to 0.0286. CLIPPEDGOSSIP and gossip averaging converge faster as the theory predicts. However, TM, GM, and MOZI are still stuck at 50% for the same heterogeneity reason.
- Bucketing and adding a random edge help all aggregators exceed 50% accuracy.

## 6.2 DECENTRALIZED LEARNING UNDER MORE ATTACKS AND TOPOLOGIES.

In this section, we compare robust aggregators over more topologies and Byzantine attacks in the non-IID setting. We consider two topologies: randomized small world ($\gamma = 0.084$) and torus ($\gamma = 0.131$). They are much less restrictive than the dumbbell topology ($\gamma = 0.043$) where all existing aggregators fail to reach consensus even $\delta = 0$. For attacks, we implement state of the art federated attacks Inner product manipulation (**IPM**) (Xie et al., 2019) and A little is enough (**ALIE**) (Baruch et al., 2019) and label-flipping (LF) and bit-flipping (BF). Details about topologies and the adaptation of FL attacks to the decentralized setup are provided in § C.1 and § B.

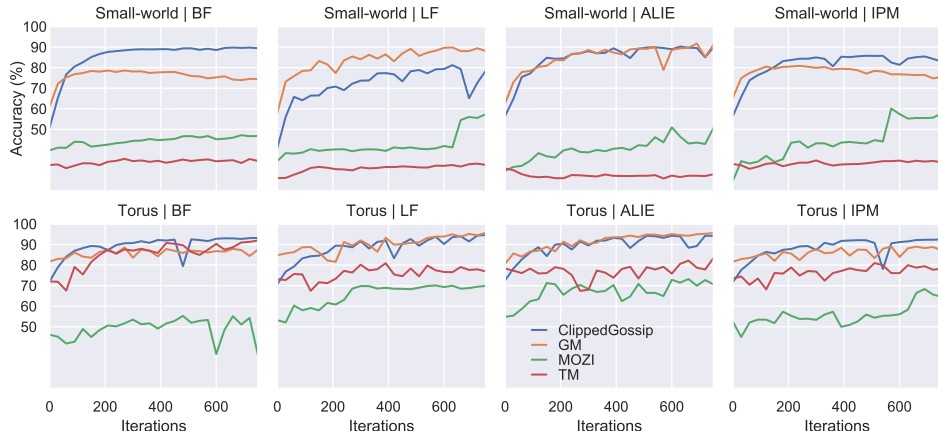

Figure 5: Robust aggregators on randomized small-world (10 regular nodes) and torus topology (9 regular nodes) under Byzantine attacks (2 attackers). We observe that across all attacks and networks, clipped gossip has excellent performance, with the geometric median (GM) coming second.

The results in Fig. 5 show that CLIPPEDGOSSIP has consistently superior performance under all topologies and attacks. All robust aggregators are generally performing better on easier topology (large $\gamma$). The GM has a very good performance on these two topologies but, as we have demonstrated in the dumbbell topology, GM does not work in more challenging topologies. Therefore, CLIPPEDGOSSIP is recommended for a general constrained topology.

### 6.3 LOWER BOUND OF OPTIMIZATION

We empirically investigate the lower bound of optimization $O(\delta_{\max}\zeta^2\gamma^{-2})$ in Theorem III. In this experiment, we fix spectral gap $\gamma$, heterogeneity $\zeta^2$ and use different $\delta_{\max}$ fractions of Byzantine edges in the dumbbell topology. The Byzantine workers are added to $V_1$ in clique A and its mirror node in clique B. We define the following *dissensus* attack for decentralized optimization

**Definition A** (DISSENSUS attack)**.** *For $i \in \mathcal{V}_{\mathrm{R}}$ and $\varepsilon_i > 0$, a dissensus attacker $j \in \mathcal{N}_i \cap \mathcal{V}_{\mathrm{B}}$ sends*

$$\boldsymbol{x}_j := \boldsymbol{x}_i - \varepsilon_i \frac{\sum_{k \in \mathcal{N}_i \cap \mathcal{V}_{\mathrm{R}}} \boldsymbol{W}_{ik}(\boldsymbol{x}_k - \boldsymbol{x}_i)}{\sum_{j \in \mathcal{N}_i \cap \mathcal{V}_{\mathrm{B}}} \boldsymbol{W}_{ij}}. \tag{6}$$

The resulting Figure 6 shows that with increasing $\delta_{\max}$ the model quality drops significantly. This is in line with our proven robust convergence rate in terms of $\delta_{\max}$. Notice that for large $\delta_{\max}$, the model averaged over all workers performs even worse than those averaged within cliques. It means the models in two cliques are essentially disconnected and are converging to different local minima or stationary points of a non-convex landscape. See § D.2.2 for details.

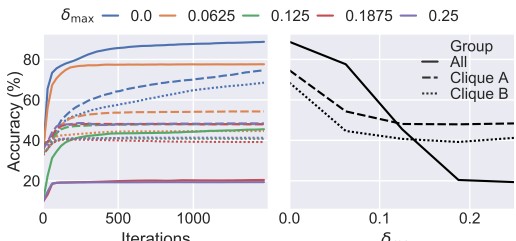

## 7 DISCUSSION

The main takeaway from our work is that ill-connected communication topologies can vastly magnify the effect of bad actors. As long as the communication topology is reasonably well connected (say $\gamma = 0.35$) and the fraction of attackers is mild (say $\delta = 10\%$), clipped gossip provably ensures robustness. Under more ex-

Figure 6: Effect of the number of attackers on the accuracy of CLIPPEDGOSSIP under dissensus attack with varying $\delta_{\max}$ and fixed $\gamma$, $\zeta^2$. The solid (resp. dashed) lines denote models averaged over all (resp. clique A or B) regular workers. The right figure shows the performance of the last iterates of curves in the left figure.

treme conditions, however, *no algorithm* can guarantee robust convergence. Given that decentralized consensus has been proposed as a backbone for digital democracy (Bulteau et al., 2021), and that decentralized learning is touted to be an alternative to current centralized training paradigms, our findings are significant. A simple strategy we recommend (along with using CLIPPEDGOSSIP) is adding random edges to improve the connectivity and robustify the network.

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

# Appendices

CONTENTS OF THE APPENDIX

## A  EXISTING ROBUST AGGREGATORS

In this section, we describe existing robust aggregators mentioned in this paper. Regular nodes can replace gossip averaging (GOSSIP) with robust aggregators in the federated learning. Let's take geometric median and trimmed mean for example.

- **Geometric median (GM).** Pillutla et al. (2019) implements the geometric median

$$\text{GM}(\boldsymbol{x}_1, \ldots, \boldsymbol{x}_n) := \arg\min_{\boldsymbol{v}} \sum_{i=1}^{n} \|\boldsymbol{v} - \boldsymbol{x}_i\|_2.$$

- **Coordinate-wise trimmed mean (TM).** Yin et al. (2018); Yang & Bajwa (2019a) computes the $k$-th coordinate of TM as

$$[\text{TM}(\boldsymbol{x}_1, \ldots, \boldsymbol{x}_n)]_k := \tfrac{1}{(1-2\beta)n} \sum_{i \in U_k} [\boldsymbol{x}_i]_k$$

where $U_k$ is a subset of $[n]$ obtained by removing the largest and smallest $\beta$-fraction of its elements.

These aggregators don't take advantage of the trusted local information and treat all models equally.

The MOZI algorithm (Guo et al., 2021) leverages local information to filter outliers.

- **Mozi.** Guo et al. (2021) applies two screening steps on worker $i \in \mathcal{V}_{\mathsf{R}}$

$$\mathcal{N}_i^s := \arg\min_{\substack{\mathcal{N}^* \subset \mathcal{N}_i \\ |\mathcal{N}^*| = \delta_i |\mathcal{N}_i|}} \sum_{j \in \mathcal{N}^*} \|\boldsymbol{x}_i - \boldsymbol{x}_j\|,$$

$$\mathcal{N}_i^r := \mathcal{N}_i^s \cap \{j \in [n] : \ell(\boldsymbol{x}_j, \xi_i) \leq \ell(\boldsymbol{x}_i, \xi_i)\}$$

where $\xi_i \sim \mathcal{D}_i$ is a random sample. If $\mathcal{N}_i^r = \emptyset$, then redefine $\mathcal{N}_i^r := \{\arg\min_j \ell(\boldsymbol{x}_j, \xi_i)\}$. Then they update the model with

$$\boldsymbol{x}_i^{t+1} := \alpha \boldsymbol{x}_i^t + \frac{1-\alpha}{|\mathcal{N}_i^r|} \sum_{j \in \mathcal{N}_i^r} \boldsymbol{x}_j^t - \eta \nabla F_i(\boldsymbol{x}_i^t; \xi_i^t)$$

where $\alpha \in [0, 1]$ is an hyperparameter.

# B  BYZANTINE ATTACKS IN THE DECENTRALIZED ENVIRONMENT

In this section, we first describe how to transform attacks from the federated learning to the decentralized environment. Then we introduce the *dissensus* attack for decentralized environment.

## B.1  EXISTING ATTACKS IN FEDERATED LEARNING

**A little is enough (ALIE).**  The attackers estimate the mean $\mu_{\mathcal{N}_i}$ and standard deviation $\sigma_{\mathcal{N}_i}$ of the regular models, and send $\mu_{\mathcal{N}_i} - z\sigma_{\mathcal{N}_i}$ to regular worker $i$ where $z$ is a small constant controlling the strength of the attack (Baruch et al., 2019). The hyperparameter $z$ for ALIE is computed according to (Baruch et al., 2019)

$$z = \max_z \left( \phi(z) < \frac{n - b - s}{n - b} \right) \tag{7}$$

where $s = \lfloor \frac{n}{2} + 1 \rfloor - b$ and $\phi$ is the cumulative standard normal function.

**Inner product manipulation attack (IPM).**  The inner product manipulation attack is proposed in (Xie et al., 2019) which lets all attackers send same corrupted gradient $\boldsymbol{u}$ based on the good gradients

$$\boldsymbol{u}_j = -\varepsilon \text{AVG}(\{\boldsymbol{v}_i : i \in \mathcal{V}_R\}) \qquad \forall j \in \mathcal{V}_B.$$

If $\varepsilon$ is small enough, then $\boldsymbol{u}_j$ can be detected as **good** by the defense, circumventing the defense. There are 3 main differences where IPM need to adapt to the decentralized environment:

1. Byzantine workers may not connected to the same good worker.
2. The model vectors are transmitted instead of gradients.
3. The AVG should be replaced by its equivalent gossip form.

This motivates our *dissensus* attack in the next section.

## B.2  DISSENSUS ATTACK AND OTHER ATTACKS IN THE DECENTRALIZED ENVIRONMENT

In this section, we introduce a novel *dissensus* attack inspired by our impossibility construction in Theorem II and the IPM attack described above. The dissensus attack aims to prevent regular worker models from reaching consensus. Roughly speaking, dissensus attackers around worker $i$ send its model weights that are symmetric to the weighted average of regular neighbors around $i$. Then after gossip averaging step, the consensus distance drops slower or even grows which motivates the name "dissensus".

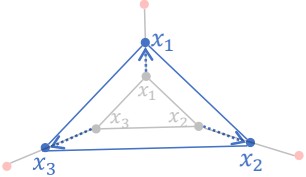

We can parameterize the attack through hyperparameter $\varepsilon_i$ and summarize the attack in Definition A

$$\boldsymbol{x}_j := \boldsymbol{x}_i - \varepsilon_i \frac{\sum_{k \in \mathcal{N}_i \cap \mathcal{V}_R} \boldsymbol{W}_{ik}(\boldsymbol{x}_k - \boldsymbol{x}_i)}{\sum_{j \in \mathcal{N}_i \cap \mathcal{V}_B} \boldsymbol{W}_{ij}}. \tag{8}$$

The $\varepsilon_i$ determines the behavior of the attack. By taking smaller $\varepsilon_i$, Byzantine model weights are closer to the target updates $i$ and difficult to be detected. On the other hand, a larger $\varepsilon_i$ pulls the model away from the consensus.

Figure 7: Example of the DISSENSUS attack. The gray (resp. red) denotes regular (resp. Byzantine) nodes. The blue dots represents the parameters of regular nodes after gossip steps.

Note that this attack requires omniscience since it exploits model information from across the network. If the attackers in addition can choose which node to attack, then they can choose either to spread about the attack across the network or focus on the targeting graph cut, that is min-cut of the graph.

**Effect of the dissensus attack.** The dissensus attack enjoy the following properties.

**Proposition IV.** (i) *For all* $i \in \mathcal{V}_R$, *under the dissensus attack with* $\varepsilon_i = 1$, *the gossip averaging step* (GOSSIP) *is equivalent to* no *communication on* $i$, $\boldsymbol{x}_i^{t+1} = \boldsymbol{x}_i^t$. *Secondly,* (ii) *If the graph is fully connected, gossip averaging recovers the correct consensus even in the presence of dissensus attack.*

The above proposition illustrates two interesting aspects of the attack. Firstly, dissensus works by negating the progress that would be made by gossip. The attack in (Peng & Ling, 2020) also satisfies this property (see Appendix for additional discussion). Secondly, it is a uniquely decentralized attack and has no effect in the centralized setting. Hence, its effect can be used to measure the additional difficulty posed due to the restricted communication topology.

*Proof.* For the first part, by definition (GOSSIP) we know that

$$\boldsymbol{x}_i^{t+1} = \sum_{j=1}^n \boldsymbol{W}_{ij}\boldsymbol{x}_j^t = \boldsymbol{x}_i^t + \sum_{j\in\mathcal{N}_i} \boldsymbol{W}_{ij}(\boldsymbol{x}_j^t - \boldsymbol{x}_i^t)$$

By setting $\varepsilon_i = 1$ in the attack (6), the second term 0 and therefore $\boldsymbol{x}_i^{t+1} = \boldsymbol{x}_i^t$. For part (ii), note that in a fully connected graph the gossip average is the same as standard average. Averaging all the perturbations introduced by the dissensus attack gives

$$-\varepsilon \sum_{i,j\in\mathcal{V}_R} W_{i,j}(\boldsymbol{x}_j^t - \boldsymbol{x}_i^t) = 0\,.$$

All terms cancel and sum to 0 by symmetry. Thus, in a fully connected graph the dissensus perturbations cancel out and the gossip average returns the correct consensus. $\square$

**Relation with zero-sum attack and dissensus.** Peng & Ling (2020) propose the "zero-sum" attack which achieves similar effects as Proposition IV part (i). This attack is defined for $j \in \mathcal{V}_B$

$$\boldsymbol{x}_j := -\frac{\sum_{k\in\mathcal{N}_i\cap\mathcal{V}_R} \boldsymbol{x}_k}{|\mathcal{N}_i\cap\mathcal{V}_B|}\,.$$

The key difference between zero-sum attack and our proposed attack is three-fold. First, zero-sum attack ensures $\sum_{j\in\mathcal{N}_i} \boldsymbol{x}_j = 0$ which means the Byzantine models have to be far away from $\boldsymbol{x}_i^t$ and therefore easy to detect. This attack pull the aggregated model to $\boldsymbol{0}$. On the other hand, our attack ensures

$$\frac{1}{\sum_{j\in\mathcal{N}_i} \boldsymbol{W}_{ij}} \sum_{j\in\mathcal{N}_i} \boldsymbol{W}_{ij}\boldsymbol{x}_j^t = \boldsymbol{x}_i^t$$

and the Byzantine updates can be very close to $\boldsymbol{x}_i^t$ and it is more difficult to be detected. Second, our proposed attack considers the gossip averaging which is prevalent in decentralized training (Koloskova et al., 2020b) while the zero-sum attack only targets simple average. Third, our attack has an additional parameter $\varepsilon$ controlling the strength of the attack with $\varepsilon > 1$ further compromise the model quality while zero-sum attack is fixed to training alone.

## C  TOPOLOGIES AND MIXING MATRICES

### C.1  CONSTRAINED TOPOLOGIES

**Topologies that do not satisfy the robust network assumption in (LeBlanc et al., 2013; Sundaram & Gharesifard, 2018; Su & Vaidya, 2016a).** The robust network assumption requires there to be at least $b + 1$ paths between any two regular workers when there are $b$ Byzantine workers in the network (LeBlanc et al., 2013; Sundaram & Gharesifard, 2018; Su & Vaidya, 2016a). The topology in Figure 8 only has 1 path between regular workers in two cliques while having 2 Byzantine workers in the network. Therefore this topology does not satisfy the robust network assumption. But the graph cut is not adjacent to the Byzantine workers and, intuitively, it would be possible for an ideal robust aggregator to help reach consensus. The experimental results are given in Appendix D.4.

**(Randomized) Small-world topology.** The small-world topology is a random graph generated with Watts-Strogatz model (Watts & Strogatz, 1998). The topology is created using NetworkX package (Hagberg et al., 2008) with 10 regular workers each connected to 2 nearest neighbors and probability of rewiring each edge as 0.15. Two additional Byzantine workers are linked to 2 random regular workers. There are 12 workers in total.

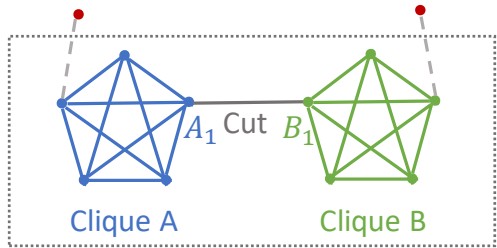

Figure 8: Example topology that does not satisfy the robust network assumptions in (Sundaram & Gharesifard, 2018; Su & Vaidya, 2016a).

**Torus topology.** The regular workers form a torus grid $T_{3,3}$ and two additional Byzantine workers are linked to 2 random regular workers. There are 11 workers in total.

The mixing matrix for these topologies are constructed with Metropolis-Hastings algorithm introduced in the previous section. The spectral gap for small-world topology and torus topology are 0.084 and 0.131 respectively. In contrast, the dumbbell topology in Figure 16 is more challenging with a spectral gap of 0.043. The data distribution is non-IID.

### C.2 CONSTRUCTING MIXING MATRICES

In this section, we introduce a few possible ways to construct the mixing weight vectors in the presence of Byzantine workers. The constructed weight vectors satisfy (A2) in Section 3.

- **Metropolis-Hastings weight (Hastings, 1970).** The Metropolis-Hastings algorithm locally constructs the mixing weights by exchanging degree information ($d_i$ and $d_j$) between two nodes $i$ and $j$. The mixing weight vector on regular worker $i \in \mathcal{V}_R$ is computed as follows

$$\boldsymbol{W}_{ij} = \begin{cases} \frac{1}{\max\{d_i, d_j\}+1} & j \in \mathcal{N}_i, \\ 1 - \sum_{l \in \mathcal{N}_i} \boldsymbol{W}_{il} & j = i, \\ 0 & \text{Otherwise.} \end{cases}$$

  If worker $j \in \mathcal{V}_B$ is Byzantine, then the only way for $j$ to maximize its weight $\boldsymbol{W}_{ij}$ to regular worker $i$ is to report a smaller degree $d_j$. However, such Byzantine behavior of node $j$ has limited influence on worker $i$'s weight $\boldsymbol{W}_{ij}$ because it can not be greater than $\frac{1}{d_i+1}$.

- **Equal-weight.** Let $d_{\max}$ be the maximum degree of nodes in a graph. Such upper bound $d_{\max}$ can be a public information, for example, a bluetooth device can at most connect to $d_{\max}$ other devices due to physical constraints. The Byzantine worker cannot change the value of $d_{\max}$. Then we use the following naive construction

$$\boldsymbol{W}_{ij} = \begin{cases} \frac{1}{d_{\max}+1} & j \in \mathcal{N}_i, \\ 1 - \frac{|\mathcal{N}_i|}{d_{\max}+1} & j = i, \\ 0 & \text{Otherwise.} \end{cases} \tag{9}$$

Note that these construction schemes are not proved to be the optimal. In this work, we focus on the Byzantine attacks given a topology and associated mixing weights. We leave it as future work to explore the best strategy to construct mixing weights.

## D EXPERIMENTS

We summarize the hardware and software for experiments in Table 2. We list the setups and results of experiments for consensus in Appendix D.1 and optimization in Appendix D.2.

### D.1 BYZANTINE-ROBUST CONSENSUS

Table 2: Runtime hardwares and softwares.

| | |
|---|---|
| **CPU** | |
| Model name | Intel (R) Xeon (R) Gold 6132 CPU @ 2.60 GHz |
| # CPU(s) | 56 |
| NUMA node(s) | 2 |
| **GPU** | |
| Product Name | Tesla V100-SXM2-32GB |
| CUDA Version | 11.0 |
| **PyTorch** | |
| Version | 1.7.1 |

Table 3: Default experimental settings for MNIST

| | |
|---|---|
| Dataset | MNIST |
| Architecture | CONV-CONV-DROPOUT-FC-DROPOUT-FC |
| Training objective | Negative log likelihood loss |
| Evaluation objective | Top-1 accuracy |
| Batch size per worker | 32 |
| Momentum | 0.9 |
| Learning rate | 0.01 |
| LR decay | No |
| LR warmup | No |
| Weight decay | No |
| Repetitions | 1 |
| Reported metric | Mean test accuracy over the last 150 iterations |

In this section, we provide detailed setups for Figure 3. The Figure 9 demonstrates the topology for the experiment. The 4 regular workers are connected with two of them holding value 0 and the others holding 200. Then the average consensus is 100 with initial mean square error equals 10000. Two Byzantine workers are connected to two regular workers in the middle. We can tune the weights of each edge to change the mixing matrix and $\gamma$. Then we can decide the weight $\delta$ on the Byzantine edge. The $\gamma$ and $\delta$ used in the experiments are

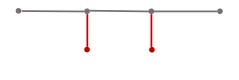

Figure 9: The topology for the attacks on consensus. The grey and red nodes denote regular and Byzantine workers respectively.

- $p := 1 - (1 - \gamma)^2 \in [0.06, 0.05, 0.04, 0.03, 0.02, 0.01, 0.005, 0.0014, 3.7e - 4, 1e - 4, 1e - 5]$
- $\delta \in [0.05, 0.1, 0.2, 0.3, 0.4, 0.5]$

where non-compatible combination of $\gamma$ and $\delta$ are ignored in the Figure 3. The dissensus attack is applied with $\varepsilon = 0.05$. The hyperparameter $\beta$ of trimmed mean (TM) is set to the actual number of Byzantine workers around the regular worker. The clipping radius of CLIPPEDGOSSIP is chosen according to (27).

In Figure 10, we show the iteration-to-error curves for all possible combinations of $\gamma$ and $\delta$. In addition, we provide a version of TM and MEDIAN which takes the mixing weight into account. As we can see, the naive TM, MEDIAN, and MEDIAN* cannot bring workers closer because of the data distribution we constructed. The TM* is performing better than the other baselines but worse than CLIPPEDGOSSIP especially on the challenging cases where $\gamma$ is small and $\delta$ is large. For CLIPPEDGOSSIP, it matches with our intuition that for a fixed $\gamma$ the convergences is worse with increasing $\delta$ while for a fixed $\delta$ the convergence is worse with decreasing $\gamma$.

### D.2 BYZANTINE-ROBUST DECENTRALIZED OPTIMIZATION

In this section, we provide detailed hyperparameters and setups for experiments in the main text and then provide additional experiments. For all MNIST tasks, we use the default setup listed in Table 3 unless specifically stated. The default hyperparameters of the robust aggregators: 1) For GM, we

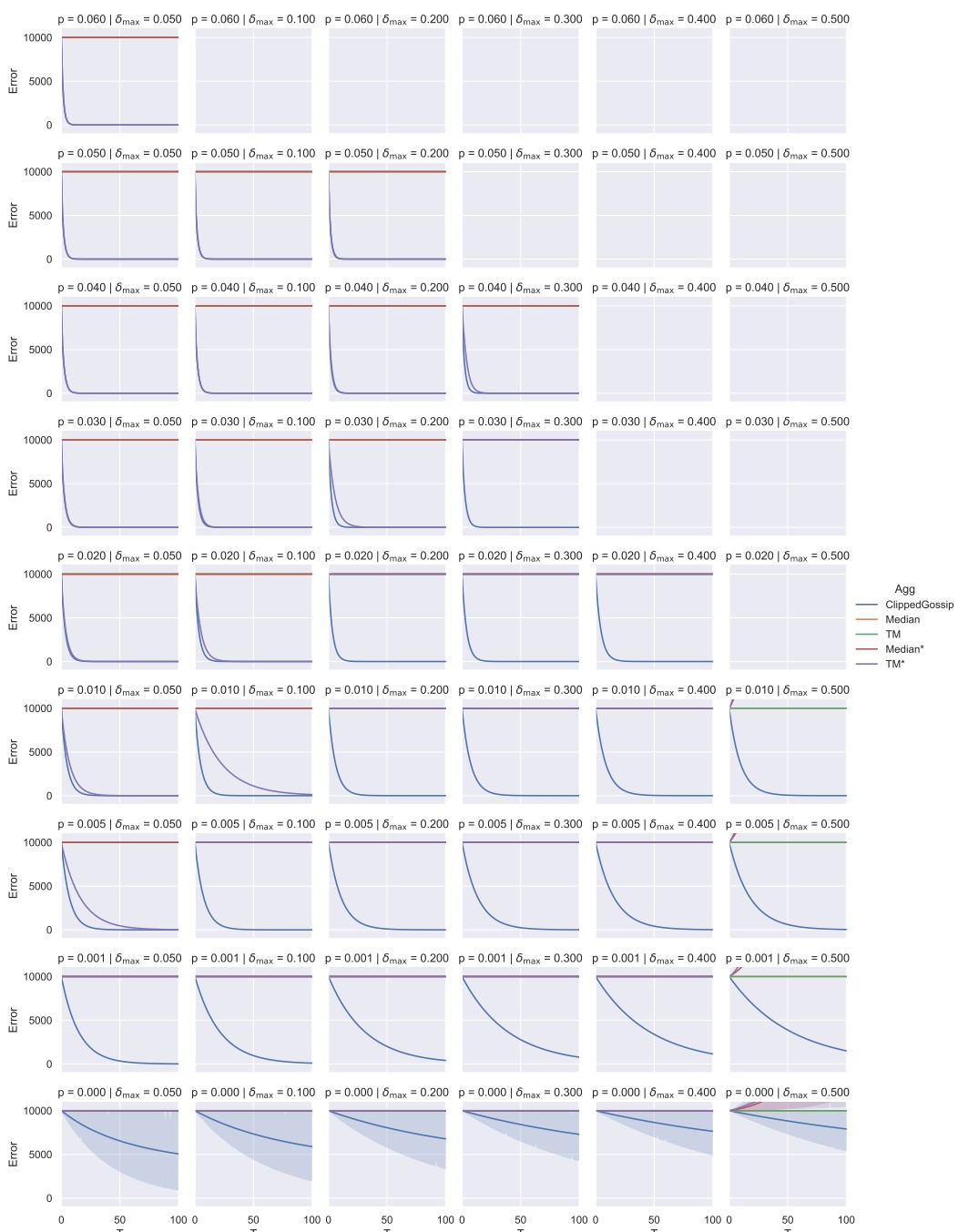

Figure 10: The iteration-to-error curves for defenses under dissensus attack. The TM* and MEDIAN* refer to the version of TM and MEDIAN which considers mixing weight.

choose number of iterations $T = 8$; 2) The TM drops top and bottom $\beta = \delta_{max} n$ percent of values in each coordinate; 3) The clipping radius of CLIPPEDGOSSIP is $\tau = 1$; 4) The model averaging hyperparameter of MOZI is $\alpha = 1$.

### D.2.1  SETUP FOR "DECENTRALIZED DEFENSES WITHOUT ATTACKERS"

The Fig. 4 uses the dumbbell topology in Fig. 1 with 10 regular workers in each clique. There is no Byzantine workers. The experiments run for 900 iterations. MOZI uses $\alpha = 0.5$ and $\rho_i = 0.99$ in

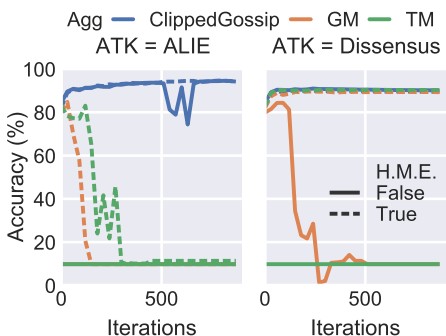

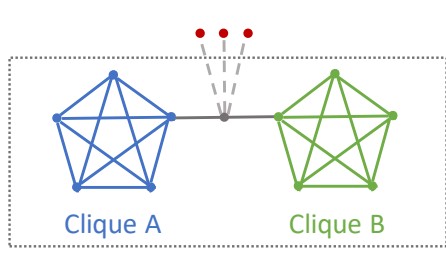
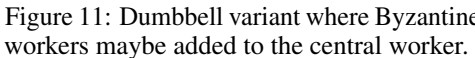

Figure 11: Dumbbell variant where Byzantine workers maybe added to the central worker.

Figure 12: Accuracy of aggregators with or without the honest majority everywhere (H.M.E.) assumption. Regular workers are connected through a ring and have IID data.

this setting. For bucketing experiment, we choose bucket size of $s = 2$. It means we randomly put at most two updates into one bucket and average within each bucket and then apply robust aggregators to the averaged updates.

### D.2.2 SETUP FOR "EFFECTS OF THE NUMBER OF BYZANTINE WORKERS"

The Fig. 6 uses a dumbbell topology variant in Fig. 11 . The experiments run for 1500 iterations. In this experiment we choose $n - b = 11$ and $b = 0, 1, 2, 3$. We choose the edge weight of Byzantine workers such that the $\widetilde{W}$ and $p$ remain the same for all these $b$. Then we can easily investigate the relation between $\delta_{\max} \in [0, \frac{b}{b+3}]$ and $p$ by varying $b$. The hyperparameter of dissensus attack is set to $\varepsilon_i = 1.5$ for all workers and all experiments.

### D.2.3 SETUP FOR "DEFENSE WITHOUT HONEST MAJORITY"

The Fig. 12 uses the ring topology of 5 regular workers in Fig. 13. 11 Byzantine workers are added to the ring so that 1 regular worker do no have honest majority. The experiments run for 900 iterations. We use $\varepsilon_i = 1.5$ for dissensus attacks. We use clipping radius $\tau = 0.1$ for CLIPPEDGOSSIP.

In the decentralized environment, the common *honest majority* assumption in the federated learning setup can be strengthen to *honest majority everywhere*, meaning all regular workers have an honest majority of neighbors (Su & Vaidya, 2016b; Yang & Bajwa, 2019b;a). Considering a ring of 5 regular workers with IID data, and adding 2 Byzantine workers to each node will still satisfy the honest majority assumption everywhere. Now adding one more Byzantine worker to a node will break the assumption.

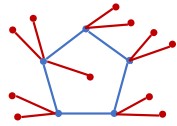

Figure 13: Ring topology without honest majority.

Figure 12 shows that while TM and GM can sometimes counter the attack under the honest majority assumption, adding one more Byzantine worker always corrupts the entire training. The CLIPPED-GOSSIP defend attacks successfully even beyond the assumption, because they leverage the fact that local updates are trustworthy. This suggest that existing statistics-based aggregators which take no advantage of local information are vulnerable under this realistic decentralized threat model.

### D.2.4 SETUP FOR "MORE TOPOLOGIES AND ATTACKS."

In Figure 5, we use the small-world and torus topologies described in Appendix C.1. More specifically, we created a randomized small-world topology using NetworkX package (Hagberg et al., 2008) with 10 regular workers each connected to 2 nearest neighbors and probability of rewiring each edge as 0.15. Two additional Byzantine workers are linked to 2 random regular workers. There are 12

Table 4: Default experimental settings for CIFAR-10

| | |
|---|---|
| Dataset | CIFAR-10 |
| Architecture | VGG-11Simonyan & Zisserman (2014) |
| Training objective | Cross entropy loss |
| Evaluation objective | Top-1 accuracy |
| Batch size per worker | 64 |
| Momentum | 0.9 |
| Learning rate | 0.1 |
| LR decay | 0.1 at epoch 80 and 120 |
| LR warmup | No |
| Weight decay | No |
| Repetitions | $1^4$ |
| Reported metric | Mean test accuracy over the last 150 iterations |

workers in total. For the torus topology, we let regular workers form a torus grid $T_{3,3}$ where all 9 regular workers are connected to 3 other workers. Two additional Byzantine workers are linked to 2 random regular workers. There are 11 workers in total.

The mixing matrix for these topologies are constructed with Metropolis-Hastings algorithm in Appendix C.2. The spectral gap for small-world topology and torus topology are 0.084 and 0.131 respectively. In contrast, the dumbbell topology in Figure 16 is more challenging with a spectral gap of 0.043. The data distribution is non-IID.

## D.3 EXPERIMENT: CIFAR-10 TASK

In this section, we conduct experiments on CIFAR-10 dataset Krizhevsky et al. (2009). The running environment of this experiment is the same as MNIST experiment Table 2. The default setup for CIFAR-10 experiment is summarized in Table 4.

We compare performances of 5 aggregators on dumbbell topology with 10 nodes in each clique (no attackers). The results of experiments are shown in Figure 14. In order to investigate if consensus has reached among the workers, we average the worker nodes in 3 different categories ( "Global", Clique A, and Clique B) and compare their performances on IID and NonIID datasets. The "IID-Global" result show that GM and TM is much worse than CLIPPEDGOSSIP and Gossip, in contrast to the MNIST experiment Figure 4 where they have matching result. This is because the workers with in each clique are converging to different stationary point — "IID-Clique A" and "IID-Clique B" show GM and TM in each clique can reach over 80% accuracy which is close to Gossip. It demonstrates that GM and TM fail to reach consensus even in this Byzantine-free case and therefore vulnerable to attacks.

The NonIID experiment also support that CLIPPEDGOSSIP perform much better than all other robust aggregators. Notice that CLIPPEDGOSSIP's "NonIID-Global" performance is better than "NonIID-Clique A" and "NonIID-Clique B" while GM and TM's result are opposite. This is because CLIPPEDGOSSIP allows effective communication in this topology and therefore clique models are close to each other in the same local minima basin such that their average (global model) is better than both of them. The GM's and TM's clique models converge to different local minima, making their averaged model underperform.

## D.4 EXPERIMENT FOR "WEAKER TOPOLOGY ASSUMPTION"

As is mentioned in Remark 1 and Appendix C.1, the topology assumption in this work is weaker than the robust network assumption in Su & Vaidya (2016a); Sundaram & Gharesifard (2018). We use the topology in Figure 8 which consists of 10 regular workers and 2 dissensus attack workers. While this topology does not satisfy the robust network assumption, it intuitively should allow communication between two cliques as no Byzantine workers are attached to the cut. However, both GM and TM will discard the graph cut due to data heterogeneity. This shows that GM and TM impede information diffusion. On the other hand, CLIPPEDGOSSIP is the only robust aggregator which help two cliques

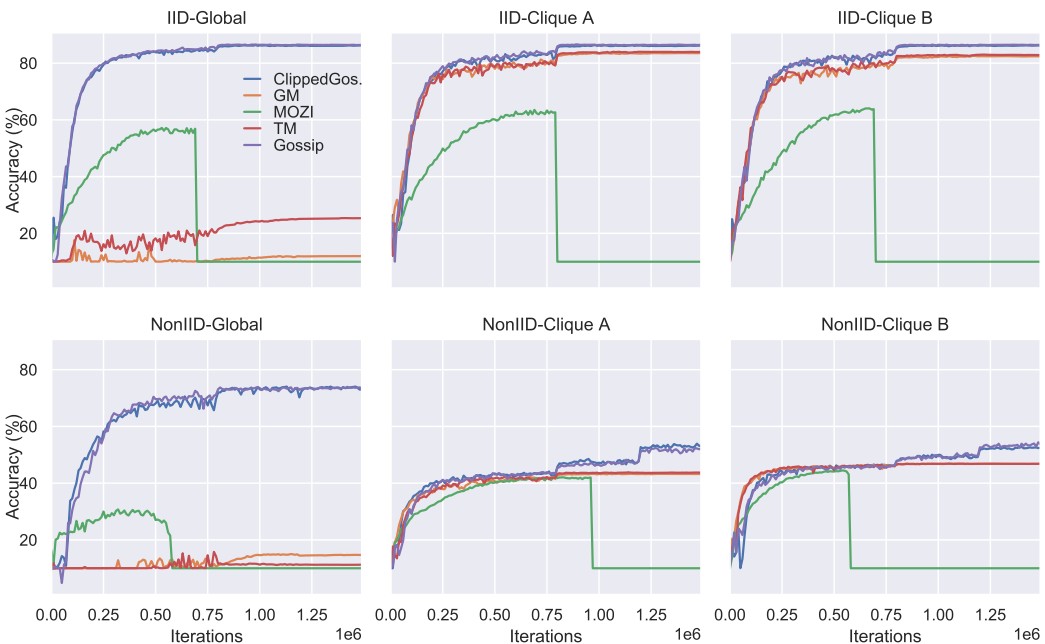

Figure 14: Train models on dumbbell topology with IID and NonIID datasets. The three figures in each row correspond to the same experiment with "Global", "Clique A", "Clique B" denoting the performances of globally averaged model, within-Clique A averaged model, and within-Clique B averaged model.

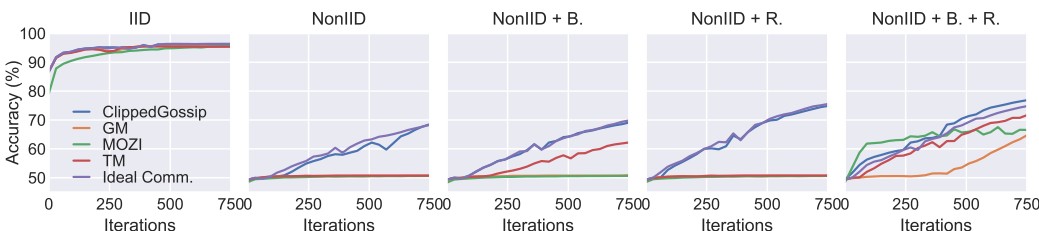

Figure 15: Compare robust aggregators under dissensus attacks over dumbbell topology Figure 5.

reaching consensus in the NonIID case. The CLIPPEDGOSSIP theoretically applies to more topologies and empirically perform better.

### D.5 EXPERIMENT: CHOOSING CLIPPING RADIUS

In Figure 16 we show the sensitive of tuning clipping radius. We use dumbbell topology with 5 regular workers in each clique and add 1 more Byzantine worker to each clique. The clipping radius is searched over a grid of $[0.1, 0.5, 1, 2, 10]$. The Byzantine workers are chosen to be Bit-Flipping, Label-Flipping, and ALIE.

We also give an adaptive clipping strategy for different $i \in \mathcal{V}_R$ and time $t$. After communication step at time $t$, the value of $\boldsymbol{x}_i^{t+1/2}$ is available. Therefore we can sort the values of $\left\| \boldsymbol{x}_i^{t+1/2} - \boldsymbol{x}_j^{t+1/2} \right\|_2^2$ for all $j \in \mathcal{N}_i$. We denote the set of indices set $\mathcal{S}_i^t$ as the indices of workers that have the smallest distances to worker $i$

$$\mathcal{S}_i^t = \underset{\mathcal{S}: \sum_{j \in \mathcal{S}} \boldsymbol{W}_{ij} \leq 1 - \delta_{\max}}{\arg\min} \sum_{j \in \mathcal{S}} \left\| \boldsymbol{x}_i^{t+1/2} - \boldsymbol{x}_j^{t+1/2} \right\|_2^2 .$$

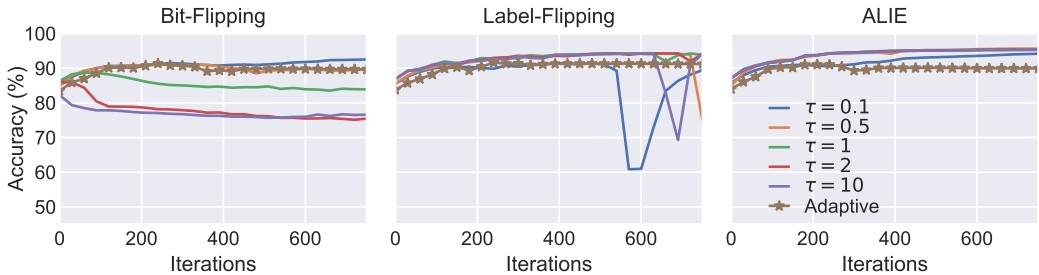

Figure 16: Tuning clipping radius on the dumbbell topology against Byzantine attacks. The y-axis is the averaged test accuracy over all of the regular workers.

Then the adaptive strategy picks clipping radius as follows

$$\tau_i^{t+1} = \sqrt{\sum_{j\in\mathcal{S}_i^t} \boldsymbol{W}_{ij} \left\|\boldsymbol{x}_i^{t+1/2} - \boldsymbol{x}_j^{t+1/2}\right\|_2^2}. \tag{10}$$

Note that this adaptive choice of clipping radius is generally a bit smaller than the theoretical value (27). It guarantees that the Byzantine workers have limited influences at cost of small slow down on the convergence.

As we can see from Figure 16, the performances of CLIPPEDGOSSIP are similar with different constant choices of $\tau$ which shows that the choice of $\tau$ is not very sensitive. The adaptive algorithms perform well in all cases. Therefore, the adaptive choice of $\tau$ will be recommended in general.

## E   ANALYSIS

We restate the core equations in Algorithm 1 at time $t$ on worker $i$ as follows

$$\boldsymbol{m}_i^{t+1} = (1-\alpha)\boldsymbol{m}_i^t + \alpha\boldsymbol{g}_i(\boldsymbol{x}_i^t) \tag{11}$$

$$\boldsymbol{x}_i^{t+1/2} = \boldsymbol{x}_i^t - \eta\boldsymbol{m}_i^{t+1} \tag{12}$$

$$\boldsymbol{z}_{j\to i}^{t+1} = \boldsymbol{x}_i^{t+1/2} + \text{CLIP}(\boldsymbol{x}_j^{t+1/2} - \boldsymbol{x}_i^{t+1/2}, \tau_i^t) \tag{13}$$

$$\boldsymbol{x}_i^{t+1} = \sum_{j=1}^n \boldsymbol{W}_{ij}\boldsymbol{z}_{j\to i}^{t+1} \tag{14}$$

In addition, we define the following virtual iterates on the set of good nodes $\mathcal{V}_R$

- $\overline{\boldsymbol{x}}^t = \frac{1}{|\mathcal{V}_R|}\sum_{i\in\mathcal{V}_R}\boldsymbol{x}_i^t$ the average (over time) of good iterates.
- $\overline{\boldsymbol{m}}^t = \frac{1}{|\mathcal{V}_R|}\sum_{i\in\mathcal{V}_R}\boldsymbol{m}_i^t$ the average (over time) of momentum iterates.

In this proof, we define $p := 1 - (1-\gamma)^2 \in (0, 1]$ for convenience.

In this section, we show that the convergence behavior of the virtual iterates $\overline{\boldsymbol{x}}^t$. The structure of this section is as follows:

- In Appendix E.1, we give common quantities, simplified notations and list common equalities/inequalities used in the proof.
- In Appendix E.2, we provide all auxiliary lemmas necessary for the proof. Among these lemmas, Lemma 8 is the key sufficient descent lemma.
- In Appendix E.3, we provide the proof of the main theorem.

### E.1   DEFINITIONS, AND INEQUALITIES

**Notations for the proof.**   We use the following variables to simplify the notation

- Optimization sub-optimality:
$$r^t := f(\bar{\boldsymbol{x}}^t) - f^\star$$

- Consensus distance:
$$\Xi^t := \frac{1}{|\mathcal{V}_{\mathsf{R}}|} \sum_{i \in \mathcal{V}_{\mathsf{R}}} \|\boldsymbol{x}_i^t - \bar{\boldsymbol{x}}^t\|_2^2$$

- The distance between the ideal gradient and actual averaged momentum
$$e_1^{t+1} := \mathbb{E}\|\nabla f(\bar{\boldsymbol{x}}^t) - \bar{\boldsymbol{m}}^{t+1}\|_2^2$$

- Similarly, the distance between the ideal gradient and individual momentums
$$\tilde{e}_1^{t+1} := \frac{1}{|\mathcal{V}_{\mathsf{R}}|} \sum_{i \in \mathcal{V}_{\mathsf{R}}} \mathbb{E}\|\nabla f(\bar{\boldsymbol{x}}^t) - \boldsymbol{m}_i^{t+1}\|_2^2$$

- Similar, distance between individual ideal gradients and individual momentums which is weighted by the mixing matrix
$$\bar{e}_1^{t+1} := \frac{1}{|\mathcal{V}_{\mathsf{R}}|} \sum_{i \in \mathcal{V}_{\mathsf{R}}} \mathbb{E}\| \sum_{j \in \mathcal{V}_{\mathsf{R}}} \widetilde{\boldsymbol{W}}_{ij}(\nabla f_j(\bar{\boldsymbol{x}}^t) - \boldsymbol{m}_j^{t+1})\|_2^2$$

- Similar we have distance between individual ideal gradients and individual momentums
$$e_{\boldsymbol{I}}^{t+1} := \frac{1}{|\mathcal{V}_{\mathsf{R}}|} \sum_{i \in \mathcal{V}_{\mathsf{R}}} \mathbb{E}\|\boldsymbol{m}_i^{t+1} - \nabla f_i(\bar{\boldsymbol{x}}^t)\|_2^2,$$

- Let $e_2^{t+1}$ be the averaged squared error introduced by clipping and Byzantine workers

$$e_2^{t+1} := \frac{1}{|\mathcal{V}_{\mathsf{R}}|} \sum_{i \in \mathcal{V}_{\mathsf{R}}} \mathbb{E} \left\| \sum_{j \in \mathcal{V}_{\mathsf{R}}} \boldsymbol{W}_{ij}(\boldsymbol{z}_{j \to i}^{t+1} - \boldsymbol{x}_j^{t+1/2}) + \sum_{j \in \mathcal{V}_{\mathsf{B}}} \boldsymbol{W}_{ij}(\boldsymbol{z}_{j \to i}^{t+1} - \boldsymbol{x}_i^{t+1/2}) \right\|_2^2.$$

**Lemma 6** (Common equalities and inequalities). *We use the following equalities and inequalities*

- *The cosine theorem:* $\forall \, \boldsymbol{x}, \boldsymbol{y} \in \mathbb{R}^d$
$$\langle \boldsymbol{x}, \boldsymbol{y} \rangle = -\frac{1}{2}\|\boldsymbol{x} - \boldsymbol{y}\|_2^2 + \frac{1}{2}\|\boldsymbol{x}\|_2^2 + \frac{1}{2}\|\boldsymbol{y}\|_2^2 \tag{15}$$

- *Young's inequality: For* $\varepsilon > 0$ *and* $\boldsymbol{x}, \boldsymbol{y} \in \mathbb{R}^d$
$$\|\boldsymbol{x} + \boldsymbol{y}\|_2^2 \le (1 + \varepsilon)\|\boldsymbol{x}\|_2^2 + (1 + \varepsilon^{-1})\|\boldsymbol{y}\|_2^2 \tag{16}$$

- *If $f$ is convex, then for* $\alpha \in [0, 1]$ *and* $\boldsymbol{x}, \boldsymbol{y} \in \mathbb{R}^d$
$$f(\alpha\boldsymbol{x} + (1 - \alpha)\boldsymbol{y}) \le \alpha f(\boldsymbol{x}) + (1 - \alpha)f(\boldsymbol{y}) \tag{17}$$

- *Cauchy-Schwarz inequality*
$$\langle \boldsymbol{x}, \boldsymbol{y} \rangle \le \|\boldsymbol{x}\|_2 \|\boldsymbol{y}\|_2 \tag{18}$$

- *Let* $\{\boldsymbol{x}_i : i \in [m]\}$ *be independent random variables and* $\mathbb{E}\,\boldsymbol{x}_i = \boldsymbol{0}$ *and* $\mathbb{E}\|\boldsymbol{x}_i\|^2 = \sigma^2$ *then*
$$\mathbb{E}\|\frac{1}{m} \sum_{i=1}^m \boldsymbol{x}_i\|_2^2 = \frac{\sigma^2}{m} \tag{19}$$

### E.2 Lemmas

The following lemma establish the update rule for $\bar{\boldsymbol{x}}^t$.

**Lemma 7.** *Assume Lemma 3. Let* $\Delta^{t+1}$ *be the error incurred by clipping and* $\mathcal{V}_{\mathsf{B}}$

$$\Delta^{t+1} := \frac{1}{|\mathcal{V}_{\mathsf{R}}|} \sum_{i \in \mathcal{V}_{\mathsf{R}}} \left( \sum_{j \in \mathcal{V}_{\mathsf{R}}} \boldsymbol{W}_{ij}(\boldsymbol{z}_{j \to i}^{t+1} - \boldsymbol{x}_j^{t+1/2}) + \sum_{j \in \mathcal{V}_{\mathsf{B}}} \boldsymbol{W}_{ij}(\boldsymbol{z}_{j \to i}^{t+1} - \boldsymbol{x}_i^{t+1/2}) \right). \tag{20}$$

*Then the virtual iterate updates*

$$\bar{\boldsymbol{x}}^{t+1} = \bar{\boldsymbol{x}}^t - \eta\bar{\boldsymbol{m}}^{t+1} + \Delta^{t+1}. \tag{21}$$

*Proof.* Expand $\bar{x}^{t+1}$ with the definition of $x_i^{t+1}$ in (14) yields

$$
\begin{aligned}
\bar{x}^{t+1} =& \frac{1}{|\mathcal{V}_\mathsf{R}|} \sum_{i \in \mathcal{V}_\mathsf{R}} x_i^{t+1} = \frac{1}{|\mathcal{V}_\mathsf{R}|} \sum_{i \in \mathcal{V}_\mathsf{R}} \left( \sum_{j \in \mathcal{V}_\mathsf{R}} W_{ij} z_{j \to i}^{t+1} + \sum_{j \in \mathcal{V}_\mathsf{B}} W_{ij} z_{j \to i}^{t+1} \right) \\
=& \frac{1}{|\mathcal{V}_\mathsf{R}|} \sum_{i \in \mathcal{V}_\mathsf{R}} \left( \sum_{j \in \mathcal{V}_\mathsf{R}} W_{ij} (z_{j \to i}^{t+1} - x_j^{t+1/2}) + \sum_{j \in \mathcal{V}_\mathsf{R}} W_{ij} x_j^{t+1/2} \right) \\
&+ \frac{1}{|\mathcal{V}_\mathsf{R}|} \sum_{i \in \mathcal{V}_\mathsf{R}} \left( \sum_{j \in \mathcal{V}_\mathsf{B}} W_{ij} (z_{j \to i}^{t+1} - x_i^{t+1/2}) + \sum_{j \in \mathcal{V}_\mathsf{B}} W_{ij} x_i^{t+1/2} \right).
\end{aligned}
$$

Reorganize the terms to form $\Delta^{t+1}$

$$
\begin{aligned}
\bar{x}^{t+1} =& \frac{1}{|\mathcal{V}_\mathsf{R}|} \sum_{i \in \mathcal{V}_\mathsf{R}} \left( \sum_{j \in \mathcal{V}_\mathsf{R}} W_{ij} x_j^{t+1/2} + \sum_{j \in \mathcal{V}_\mathsf{B}} W_{ij} x_i^{t+1/2} \right) + \Delta^{t+1} \\
=& \frac{1}{|\mathcal{V}_\mathsf{R}|} \sum_{j \in \mathcal{V}_\mathsf{R}} (1 - \delta_j) x_j^{t+1/2} + \frac{1}{|\mathcal{V}_\mathsf{R}|} \sum_{i \in \mathcal{V}_\mathsf{R}} \delta_i x_i^{t+1/2} + \Delta^{t+1} \\
=& \frac{1}{|\mathcal{V}_\mathsf{R}|} \sum_{i \in \mathcal{V}_\mathsf{R}} x_i^{t+1/2} + \Delta^{t+1} = \frac{1}{|\mathcal{V}_\mathsf{R}|} \sum_{i \in \mathcal{V}_\mathsf{R}} (x_i^t - \eta m_i^{t+1}) + \Delta^{t+1} \\
=& \bar{x}_i^t - \eta \bar{m}^{t+1} + \Delta^{t+1}. \qquad\qquad\qquad\qquad\qquad\qquad\qquad \square
\end{aligned}
$$

Note that the $\Delta^{t+1}$ can be written as the follows

$$
\Delta^{t+1} = \frac{1}{|\mathcal{V}_\mathsf{R}|} \sum_{i \in \mathcal{V}_\mathsf{R}} \left( x_i^{t+1} - \sum_{j \in \mathcal{V}_\mathsf{R}} \tilde{W}_{ij} x_j^{t+1/2} \right) = \bar{x}^{t+1} - \frac{1}{|\mathcal{V}_\mathsf{R}|} \sum_{i \in \mathcal{V}_\mathsf{R}} x_i^{t+1/2}.
$$

where measures the error introduced to $\bar{x}^{t+1}$ considering the impact of Byzantine workers and clipping. Therefore when $\mathcal{V}_\mathsf{B} = \emptyset$ and $\tau$ is sufficiently large, $\Delta^{t+1} = 0$ and $\bar{x}^{t+1}$ converge at the same rate as the centralized SGD with momentum.

Recall that $e_1^{t+1} := \mathbb{E}\|\nabla f(\bar{x}^t) - \bar{m}^{t+1}\|_2^2$. The key descent lemma is stated as follow

**Lemma 8** (Sufficient decrease). *Assume (A4) and $\eta \leq \frac{1}{2L}$, then*

$$
\mathbb{E} f(\bar{x}^{t+1}) \leq f(\bar{x}^t) - \frac{\eta}{2}\|\nabla f(\bar{x}^t)\|_2^2 - \frac{\eta}{4}\mathbb{E}\|\bar{m}^{t+1} - \frac{1}{\eta}\Delta^{t+1}\|_2^2 + \eta e_1^{t+1} + \frac{1}{\eta} e_2^{t+1}.
$$

*Proof.* Use smoothness (A4) and expand it with (21)

$$
f(\bar{x}^{t+1}) \leq f(\bar{x}^t) - \langle \nabla f(\bar{x}^t), \eta \bar{m}^{t+1} - \Delta^{t+1} \rangle + \frac{L}{2}\|\eta \bar{m}^{t+1} - \Delta^{t+1}\|_2^2
$$

Apply cosine theorem (15) to the inner product $\eta \langle \nabla f(\bar{x}^t), \bar{m}^{t+1} - \frac{1}{\eta}\Delta^{t+1} \rangle$ yields

$$
\begin{aligned}
\mathbb{E} f(\bar{x}^{t+1}) \leq& f(\bar{x}^t) - \frac{\eta}{2}\|\nabla f(\bar{x}^t)\|_2^2 - \left( \frac{\eta - L\eta^2}{2} \right) \mathbb{E}\|\bar{m}^{t+1} - \frac{1}{\eta}\Delta^{t+1}\|_2^2 \\
&+ \frac{\eta}{2}\mathbb{E}\|\nabla f(\bar{x}^t) - \bar{m}^{t+1} + \frac{1}{\eta}\Delta^{t+1}\|_2^2.
\end{aligned}
$$

If step size $\eta \leq \frac{1}{2L}$, then $-\frac{\eta - L\eta^2}{2} \leq -\frac{\eta}{4}$. Applying inequality (16) to the last term

$$
\frac{\eta}{2}\mathbb{E}\|\nabla f(\bar{x}^t) - \bar{m}^{t+1} + \frac{1}{\eta}\Delta^{t+1}\|_2^2 \leq \eta \mathbb{E}\|\nabla f(\bar{x}^t) - \bar{m}^{t+1}\|_2^2 + \frac{1}{\eta}\mathbb{E}\|\Delta^{t+1}\|_2^2.
$$

Since $e_1^{t+1} := \mathbb{E}\|\nabla f(\bar{x}^t) - \bar{m}^{t+1}\|_2^2$ and $\mathbb{E}\|\Delta^{t+1}\|_2^2 \leq e_2^{t+1}$, then we have

$$
\mathbb{E} f(\bar{x}^{t+1}) \leq f(\bar{x}^t) - \frac{\eta}{2}\|\nabla f(\bar{x}^t)\|_2^2 - \frac{\eta}{4}\mathbb{E}\|\bar{m}^{t+1} - \frac{1}{\eta}\Delta^{t+1}\|_2^2 + \eta e_1^{t+1} + \frac{1}{\eta} e_2^{t+1}. \qquad \square
$$

In the next lemma, we establish the recursion for the distance between momentums and gradients

**Lemma 9.** *Assume (A3) and (A4) and Lemma 3, For any doubly stochastic mixing matrix $\boldsymbol{A} \in \mathbb{R}^{n \times n}$*

$$e_A^{t+1} = \frac{1}{|\mathcal{V}_{\mathsf{R}}|} \sum_{i \in \mathcal{V}_{\mathsf{R}}} \mathbb{E}\|\sum_{j \in \mathcal{V}_{\mathsf{R}}} \boldsymbol{A}_{ij}(\boldsymbol{m}_j^{t+1} - \nabla f_j(\bar{\boldsymbol{x}}^t))\|_2^2,$$

*then we have the following recursion*

$$e_A^{t+1} \leq (1-\alpha)e_A^t + \frac{\alpha^2\sigma^2}{|\mathcal{V}_{\mathsf{R}}|}\|\boldsymbol{A}\|_{F,\mathcal{V}_{\mathsf{R}}}^2 + 2\alpha L^2 \Xi^t + \frac{2L^2\eta^2}{\alpha}\|\bar{\boldsymbol{m}}^t - \frac{1}{\eta}\Delta^t\|_2^2. \tag{22}$$

*where we define $\|\boldsymbol{A}\|_{F,\mathcal{V}_{\mathsf{R}}}^2 := \sum_{i \in \mathcal{V}_{\mathsf{R}}} \sum_{j \in \mathcal{V}_{\mathsf{R}}} \boldsymbol{A}_{ij}^2$ Therefore,*

- *If $\boldsymbol{A}_{ij} = \frac{1}{|\mathcal{V}_{\mathsf{R}}|}$ for all $i,j \in \mathcal{V}_{\mathsf{R}}$, then $e_A^{t+1} = e_1^{t+1}$ and $\|\boldsymbol{A}\|_{F,\mathcal{V}_{\mathsf{R}}}^2 = 1$.*

- *If $\boldsymbol{A} = \widetilde{\boldsymbol{W}}$, then $e_A^{t+1} = \bar{e}_1^{t+1}$ and $\|\boldsymbol{A}\|_{F,\mathcal{V}_{\mathsf{R}}}^2 = \sum_{i \in \mathcal{V}_{\mathsf{R}}} \sum_{j \in \mathcal{V}_{\mathsf{R}}} \widetilde{\boldsymbol{W}}_{ij}^2 \leq |\mathcal{V}_{\mathsf{R}}|$.*

- *If $\boldsymbol{A} = \boldsymbol{I}$, then $\|\boldsymbol{A}\|_{F,\mathcal{V}_{\mathsf{R}}}^2 = |\mathcal{V}_{\mathsf{R}}|$. In addition,*

$$\tilde{e}_1^{t+1} \leq 2e_{\boldsymbol{I}}^{t+1} + 2\zeta^2$$

  *where $\boldsymbol{A} = \boldsymbol{I}$.*

*Proof.* We can expand $e_A^{t+1}$ by expanding $\boldsymbol{m}_j^{t+1}$

$$e_A^{t+1} \stackrel{(11)}{=} \frac{1}{|\mathcal{V}_{\mathsf{R}}|} \sum_{i \in \mathcal{V}_{\mathsf{R}}} \mathbb{E}\|\sum_{j \in \mathcal{V}_{\mathsf{R}}} \boldsymbol{A}_{ij}((1-\alpha)\boldsymbol{m}_j^t + \alpha\boldsymbol{g}_j(\boldsymbol{x}_j^t) - \nabla f_j(\bar{\boldsymbol{x}}^t))\|_2^2$$

$$= \frac{1}{|\mathcal{V}_{\mathsf{R}}|} \sum_{i \in \mathcal{V}_{\mathsf{R}}} \mathbb{E}\|\sum_{j \in \mathcal{V}_{\mathsf{R}}} \boldsymbol{A}_{ij}((1-\alpha)\boldsymbol{m}_j^t + \alpha(\boldsymbol{g}_j(\boldsymbol{x}_j^t) \pm \nabla f_j(\boldsymbol{x}_j^t)) - \nabla f_j(\bar{\boldsymbol{x}}^t))\|_2^2$$

Extract the stochastic term $\boldsymbol{g}_j(\boldsymbol{x}_j^t) - \nabla f_j(\boldsymbol{x}_j^t)$ inside the norm and use that $\mathbb{E}\,\boldsymbol{g}_j(\boldsymbol{x}_j^t) = \nabla f_j(\boldsymbol{x}_j^t)$,

$$e_A^{t+1} = \frac{1}{|\mathcal{V}_{\mathsf{R}}|} \sum_{i \in \mathcal{V}_{\mathsf{R}}} \|\sum_{j \in \mathcal{V}_{\mathsf{R}}} \boldsymbol{A}_{ij}((1-\alpha)\boldsymbol{m}_j^t + \alpha\nabla f_j(\boldsymbol{x}_j^t) - \nabla f_j(\bar{\boldsymbol{x}}^t))\|_2^2$$

$$+ \frac{1}{|\mathcal{V}_{\mathsf{R}}|} \sum_{i \in \mathcal{V}_{\mathsf{R}}} \mathbb{E}\|\sum_{j \in \mathcal{V}_{\mathsf{R}}} \boldsymbol{A}_{ij}\alpha(\boldsymbol{g}_j(\boldsymbol{x}_j^t) - \nabla f_j(\boldsymbol{x}_j^t))\|_2^2$$

$$\leq \frac{1}{|\mathcal{V}_{\mathsf{R}}|} \sum_{i \in \mathcal{V}_{\mathsf{R}}} \|\sum_{j \in \mathcal{V}_{\mathsf{R}}} \boldsymbol{A}_{ij}((1-\alpha)\boldsymbol{m}_j^t + \alpha\nabla f_j(\boldsymbol{x}_j^t) - \nabla f_j(\bar{\boldsymbol{x}}^t))\|_2^2$$

$$+ \frac{\alpha^2}{|\mathcal{V}_{\mathsf{R}}|} \sum_{i \in \mathcal{V}_{\mathsf{R}}} \sum_{j \in \mathcal{V}_{\mathsf{R}}} \boldsymbol{A}_{ij}^2 \,\mathbb{E}\|\boldsymbol{g}_j(\boldsymbol{x}_j^t) - \nabla f_j(\boldsymbol{x}_j^t)\|_2^2.$$

Then we can use (A3) for the last term to get

$$e_A^{t+1} = \frac{1}{|\mathcal{V}_{\mathsf{R}}|} \sum_{i \in \mathcal{V}_{\mathsf{R}}} \|\sum_{j \in \mathcal{V}_{\mathsf{R}}} \boldsymbol{A}_{ij}((1-\alpha)\boldsymbol{m}_j^t + \alpha\nabla f_j(\boldsymbol{x}_j^t) - \nabla f_j(\bar{\boldsymbol{x}}^t))\|_2^2 + \frac{\alpha^2\sigma^2}{|\mathcal{V}_{\mathsf{R}}|}\|\boldsymbol{A}\|_{F,\mathcal{V}_{\mathsf{R}}}^2.$$

Then we insert $\pm(1-\alpha)\nabla f_j(\bar{\boldsymbol{x}}^{t-1})$ inside the first norm and expand using (17)

$$e_A^{t+1} \leq \frac{1-\alpha}{|\mathcal{V}_{\mathsf{R}}|} \sum_{i \in \mathcal{V}_{\mathsf{R}}} \|\sum_{j \in \mathcal{V}_{\mathsf{R}}} \boldsymbol{A}_{ij}(\boldsymbol{m}_j^t - \nabla f_j(\bar{\boldsymbol{x}}^{t-1}))\|_2^2 + \frac{\alpha^2\sigma^2}{|\mathcal{V}_{\mathsf{R}}|}\|\boldsymbol{A}\|_{F,\mathcal{V}_{\mathsf{R}}}^2$$

$$+ \frac{\alpha}{|\mathcal{V}_{\mathsf{R}}|} \sum_{i \in \mathcal{V}_{\mathsf{R}}} \|\sum_{j \in \mathcal{V}_{\mathsf{R}}} \boldsymbol{A}_{ij}(\nabla f_j(\boldsymbol{x}_j^t) - \nabla f_j(\bar{\boldsymbol{x}}^t) + \frac{1-\alpha}{\alpha}(\nabla f_j(\bar{\boldsymbol{x}}^{t-1}) - \nabla f_j(\bar{\boldsymbol{x}}^t))\|_2^2.$$

Note that the first term is $e_A^t$ and by the convexity of $\|\cdot\|$ for the last term we have

$$
\begin{aligned}
e_A^{t+1} \leq & (1-\alpha)e_A^t + \frac{\alpha^2\sigma^2}{|\mathcal{V}_\mathsf{R}|}\|\boldsymbol{A}\|_{F,\mathcal{V}_\mathsf{R}}^2 \\
& + \frac{\alpha}{|\mathcal{V}_\mathsf{R}|}\sum_{j\in\mathcal{V}_\mathsf{R}}\|\nabla f_j(\boldsymbol{x}_j^t) - \nabla f_j(\bar{\boldsymbol{x}}^t) + \frac{1-\alpha}{\alpha}(\nabla f_j(\bar{\boldsymbol{x}}^{t-1}) - \nabla f_j(\bar{\boldsymbol{x}}^t))\|_2^2.
\end{aligned}
$$

Then we can further expand the last term

$$
\begin{aligned}
e_A^{t+1} \leq & (1-\alpha)e_A^t + \frac{\alpha^2\sigma^2}{|\mathcal{V}_\mathsf{R}|}\|\boldsymbol{A}\|_{F,\mathcal{V}_\mathsf{R}}^2 \\
& + \frac{2\alpha}{|\mathcal{V}_\mathsf{R}|}\sum_{j\in\mathcal{V}_\mathsf{R}}\|\nabla f_j(\boldsymbol{x}_j^t) - \nabla f_j(\bar{\boldsymbol{x}}^t)\|_2^2 + \frac{2(1-\alpha)^2}{\alpha|\mathcal{V}_\mathsf{R}|}\sum_{j\in\mathcal{V}_\mathsf{R}}\|\nabla f_j(\bar{\boldsymbol{x}}^{t-1}) - \nabla f_j(\bar{\boldsymbol{x}}^t)\|_2^2.
\end{aligned}
$$

Then we can apply smoothness (A4) and use $(1-\alpha)^2 \leq 1$

$$
e_A^{t+1} \leq (1-\alpha)e_A^t + \frac{\alpha^2\sigma^2}{|\mathcal{V}_\mathsf{R}|}\|\boldsymbol{A}\|_{F,\mathcal{V}_\mathsf{R}}^2 + 2\alpha L^2\Xi^t + \frac{2L^2\eta^2}{\alpha}\|\bar{\boldsymbol{m}}^t - \frac{1}{\eta}\Delta^t\|_2^2.
$$

Besides, consider $\tilde{e}_1^{t+1}$

$$
\begin{aligned}
\tilde{e}_1^{t+1} = & \frac{1}{|\mathcal{V}_\mathsf{R}|}\sum_{i\in\mathcal{V}_\mathsf{R}}\mathbb{E}\|\boldsymbol{m}_i^{t+1} - \nabla f(\bar{\boldsymbol{x}}^t)\|_2^2 = \frac{1}{|\mathcal{V}_\mathsf{R}|}\sum_{i\in\mathcal{V}_\mathsf{R}}\mathbb{E}\|\boldsymbol{m}_i^{t+1} \pm \nabla f_i(\bar{\boldsymbol{x}}^t) - \nabla f(\bar{\boldsymbol{x}}^t)\|_2^2 \\
\leq & 2\frac{1}{|\mathcal{V}_\mathsf{R}|}\sum_{i\in\mathcal{V}_\mathsf{R}}\mathbb{E}\|\boldsymbol{m}_i^{t+1} - \nabla f_i(\bar{\boldsymbol{x}}^t)\|_2^2 + 2\frac{1}{|\mathcal{V}_\mathsf{R}|}\sum_{i\in\mathcal{V}_\mathsf{R}}\|\nabla f_i(\bar{\boldsymbol{x}}^t) - \nabla f(\bar{\boldsymbol{x}}^t)\|_2^2 \\
= & 2e_I^{t+1} + 2\zeta^2.
\end{aligned}
$$

$\square$

As we know that $\|\Delta^{t+1}\|_2^2 \leq e_2^{t+1}$, then we need to finally bound $e_2^{t+1}$

**Lemma 10** (Bound on $e_2^{t+1}$). *For $\delta_{\max} := \max_{i\in\mathcal{V}_\mathsf{R}}\delta_i$, if*

$$
\tau_i^{t+1} = \sqrt{\frac{1}{\delta_i}\sum_{j\in\mathcal{V}_\mathsf{R}}\boldsymbol{W}_{ij}\,\mathbb{E}\left\|\boldsymbol{x}_i^{t+1/2} - \boldsymbol{x}_j^{t+1/2}\right\|_2^2},
$$

*then we have*

$$
e_2^{t+1} \leq c_1\delta_{\max}(2\eta^2(e_I^{t+1} + \zeta^2) + \Xi^t).
$$

*where constant $c_1 = 32$.*

*Proof.* Use Young's inequality (16) to bound $e_2^{t+1}$ by two parts

$$
\begin{aligned}
e_2^{t+1} = & \frac{1}{|\mathcal{V}_\mathsf{R}|}\sum_{i\in\mathcal{V}_\mathsf{R}}\mathbb{E}\left\|\sum_{j\in\mathcal{V}_\mathsf{R}}\boldsymbol{W}_{ij}(\boldsymbol{z}_{j\to i}^{t+1} - \boldsymbol{x}_j^{t+1/2}) + \sum_{j\in\mathcal{V}_\mathsf{B}}\boldsymbol{W}_{ij}(\boldsymbol{z}_{j\to i}^{t+1} - \boldsymbol{x}_i^{t+1/2})\right\|_2^2 \\
\leq & \underbrace{\frac{2}{|\mathcal{V}_\mathsf{R}|}\sum_{i\in\mathcal{V}_\mathsf{R}}\mathbb{E}\left\|\sum_{j\in\mathcal{V}_\mathsf{R}}\boldsymbol{W}_{ij}(\boldsymbol{z}_{j\to i}^{t+1} - \boldsymbol{x}_j^{t+1/2})\right\|_2^2}_{=:A_1} + \underbrace{\frac{2}{|\mathcal{V}_\mathsf{R}|}\sum_{i\in\mathcal{V}_\mathsf{R}}\mathbb{E}\left\|\sum_{j\in\mathcal{V}_\mathsf{B}}\boldsymbol{W}_{ij}(\boldsymbol{z}_{j\to i}^{t+1} - \boldsymbol{x}_i^{t+1/2})\right\|_2^2}_{=:A_2}.
\end{aligned}
$$

Look at the first term use triangular inequality of $\|\cdot\|$ and the definition of $\tau_i^{t+1}$

$$
\begin{aligned}
A_1 \leq & \frac{2}{|\mathcal{V}_\mathsf{R}|}\sum_{i\in\mathcal{V}_\mathsf{R}}\left(\sum_{j\in\mathcal{V}_\mathsf{R}}\boldsymbol{W}_{ij}\,\mathbb{E}\left\|\boldsymbol{z}_{j\to i}^{t+1} - \boldsymbol{x}_j^{t+1/2}\right\|_2\right)^2 \\
\leq & \frac{2}{|\mathcal{V}_\mathsf{R}|}\sum_{i\in\mathcal{V}_\mathsf{R}}\left(\frac{1}{\tau_i^{t+1}}\sum_{j\in\mathcal{V}_\mathsf{R}}\boldsymbol{W}_{ij}\,\mathbb{E}\left\|\boldsymbol{x}_i^{t+1/2} - \boldsymbol{x}_j^{t+1/2}\right\|_2^2\right)^2.
\end{aligned}
$$

The second inequality holds true because we can consider two cases of $\boldsymbol{z}_{j\to i}^{t+1}$ for all $j \in \mathcal{V}_\mathsf{R}$

- If $\|\boldsymbol{x}_i^{t+1/2} - \boldsymbol{x}_j^{t+1/2}\|_2^2 \leq \tau_i^{t+1}$, then CLIP has no effect and therefore $\boldsymbol{z}_{j\to i}^{t+1} = \boldsymbol{x}_j^{t+1/2}$

$$0 = \|\boldsymbol{z}_{j\to i}^{t+1} - \boldsymbol{x}_j^{t+1/2}\|_2 \leq \frac{1}{\tau_i^{t+1}}\|\boldsymbol{x}_i^{t+1/2} - \boldsymbol{x}_j^{t+1/2}\|_2^2.$$

- If $\|\boldsymbol{x}_i^{t+1/2} - \boldsymbol{x}_j^{t+1/2}\|_2^2 > \tau_i^{t+1}$, then $\boldsymbol{z}_{j\to i}^{t+1}$ sits between $\boldsymbol{x}_j^{t+1/2}$ and $\boldsymbol{x}_i^{t+1/2}$ with

$$\|\boldsymbol{z}_{j\to i}^{t+1} - \boldsymbol{x}_j^{t+1/2}\|_2 + \tau_i^{t+1} = \|\boldsymbol{x}_i^{t+1/2} - \boldsymbol{x}_j^{t+1/2}\|_2.$$

Therefore, using the inequality $a - \tau \leq \frac{a^2}{\tau}$ for $a > 0$ we have that

$$\|\boldsymbol{z}_{j\to i}^{t+1} - \boldsymbol{x}_j^{t+1/2}\|_2 = \|\boldsymbol{x}_i^{t+1/2} - \boldsymbol{x}_j^{t+1/2}\|_2 - \tau_i^{t+1} \leq \frac{1}{\tau_i^{t+1}}\|\boldsymbol{x}_i^{t+1/2} - \boldsymbol{x}_j^{t+1/2}\|_2^2.$$

Therefore we justify the second inequality.

On the other hand,

$$A_2 \leq \frac{2}{|\mathcal{V}_\mathsf{R}|}\sum_{i\in\mathcal{V}_\mathsf{R}}\left(\sum_{j\in\mathcal{V}_\mathsf{B}}\boldsymbol{W}_{ij}\,\mathbb{E}\left\|\boldsymbol{z}_{j\to i}^{t+1} - \boldsymbol{x}_i^{t+1/2}\right\|_2\right)^2 \leq \frac{2}{|\mathcal{V}_\mathsf{R}|}\sum_{i\in\mathcal{V}_\mathsf{R}}\left(\sum_{j\in\mathcal{V}_\mathsf{B}}\boldsymbol{W}_{ij}(\tau_i^{t+1})\right)^2$$

$$= \frac{2}{|\mathcal{V}_\mathsf{R}|}\sum_{i\in\mathcal{V}_\mathsf{R}}\delta_i^2(\tau_i^{t+1})^2.$$

Then minimizing the RHS of $e_2^{t+1}$ by tuning radius for clipping

$$\tau_i^{t+1} = \sqrt{\frac{1}{\delta_i}\sum_{j\in\mathcal{V}_\mathsf{R}}\boldsymbol{W}_{ij}\,\mathbb{E}\left\|\boldsymbol{x}_i^{t+1/2} - \boldsymbol{x}_j^{t+1/2}\right\|_2^2}$$

Then we come to the following bound

$$e_2^{t+1} \leq \frac{4}{|\mathcal{V}_\mathsf{R}|}\sum_{i\in\mathcal{V}_\mathsf{R}}\delta_i\sum_{j\in\mathcal{V}_\mathsf{R}}\boldsymbol{W}_{ij}\,\mathbb{E}\left\|\boldsymbol{x}_i^{t+1/2} - \boldsymbol{x}_j^{t+1/2}\right\|_2^2.$$

Then we expand the norm as follows

$$\begin{aligned}
\mathbb{E}\left\|\boldsymbol{x}_i^{t+1/2} - \boldsymbol{x}_j^{t+1/2}\right\|_2^2 &= \mathbb{E}\left\|\boldsymbol{x}_i^t - \eta\boldsymbol{m}_i^{t+1} - \boldsymbol{x}_j^t + \eta\boldsymbol{m}_j^{t+1}\right\|_2^2 \\
&= \mathbb{E}\left\|\boldsymbol{x}_i^t \pm \bar{\boldsymbol{x}}^t - \boldsymbol{x}_j^t + \eta\boldsymbol{m}_j^{t+1} \pm \eta\nabla f(\bar{\boldsymbol{x}}^t) - \eta\boldsymbol{m}_i^{t+1}\right\|_2^2 \\
&\leq 4\eta^2\,\mathbb{E}\|\boldsymbol{m}_i^{t+1} - \nabla f(\bar{\boldsymbol{x}}^t)\|_2^2 + 4\eta^2\,\mathbb{E}\|\boldsymbol{m}_j^{t+1} - \nabla f(\bar{\boldsymbol{x}}^t)\|_2^2 \\
&\quad + 4\|\boldsymbol{x}_i^t - \bar{\boldsymbol{x}}^t\|_2^2 + 4\|\boldsymbol{x}_j^t - \bar{\boldsymbol{x}}^t\|_2^2
\end{aligned} \tag{23}$$

Use the fact that $\sum_{j\in\mathcal{V}_\mathsf{R}}\boldsymbol{W}_{ij} = 1 - \delta_i$ we have

$$\begin{aligned}
e_2^{t+1} &\leq \frac{16\eta^2}{|\mathcal{V}_\mathsf{R}|}\sum_{i\in\mathcal{V}_\mathsf{R}}\delta_i(1-\delta_i)\,\mathbb{E}\|\boldsymbol{m}_i^{t+1} - \nabla f(\bar{\boldsymbol{x}}^t)\|_2^2 + \frac{16\eta^2}{|\mathcal{V}_\mathsf{R}|}\sum_{j\in\mathcal{V}_\mathsf{R}}\sum_{i\in\mathcal{V}_\mathsf{R}}\delta_i\boldsymbol{W}_{ij}\,\mathbb{E}\|\boldsymbol{m}_j^{t+1} - \nabla f(\bar{\boldsymbol{x}}^t)\|_2^2 \\
&\quad + \frac{16}{|\mathcal{V}_\mathsf{R}|}\sum_{i\in\mathcal{V}_\mathsf{R}}\delta_i(1-\delta_i)\|\boldsymbol{x}_i^t - \bar{\boldsymbol{x}}^t\|_2^2 + \frac{16}{|\mathcal{V}_\mathsf{R}|}\sum_{j\in\mathcal{V}_\mathsf{R}}\sum_{i\in\mathcal{V}_\mathsf{R}}\delta_i\boldsymbol{W}_{ij}\|\boldsymbol{x}_j^t - \bar{\boldsymbol{x}}^t\|_2^2
\end{aligned}$$

Use the fact that $\delta_i \leq \delta_{\max}$ and $1 - \delta_i \leq 1$ for all $i \in \mathcal{V}_\mathsf{R}$,

$$e_2^{t+1} \leq 32\delta_{\max}(2\eta^2(e_{\boldsymbol{I}}^{t+1} + \zeta^2) + \Xi^t).$$

$\square$

**Theorem I′.** *Let $\bar{\boldsymbol{x}} := \frac{1}{|\mathcal{V}_\mathsf{R}|}\sum_{i\in\mathcal{V}_\mathsf{R}}\boldsymbol{x}_i$ be the average iterate over the unknown set of regular nodes with*

$$\tau_i = \sqrt{\frac{1}{\delta_i}\sum_{j\in\mathcal{V}_\mathsf{R}}\boldsymbol{W}_{ij}\,\mathbb{E}\|\boldsymbol{x}_i - \boldsymbol{x}_j\|_2^2}. \tag{24}$$

*If the initial consensus distance is bounded as $\frac{1}{|\mathcal{V}_\mathsf{R}|}\sum_{i\in\mathcal{V}_\mathsf{R}}\mathbb{E}\|\boldsymbol{x}_i - \bar{\boldsymbol{x}}\|^2 \leq \rho^2$, then for all $i \in \mathcal{V}_\mathsf{R}$, the output $\hat{\boldsymbol{x}}_i$ of CLIPPEDGOSSIP satisfies*

$$\frac{1}{|\mathcal{V}_\mathsf{R}|}\sum_{i\in\mathcal{V}_\mathsf{R}}\mathbb{E}\|\hat{\boldsymbol{x}}_i - \bar{\boldsymbol{x}}\|^2 \leq \left(1 - \gamma + c\sqrt{\delta_{\max}}\right)^2\rho^2$$

*where the expectation is over the random variable $\{\boldsymbol{x}_i\}_{i\in\mathcal{V}_\mathsf{R}}$ and $c > 0$ is a constant.*

*Proof.* We can consider the 1-step consensus problem as 1-step of optimization problem with $\rho^2 = \Xi^t$ and $\eta = 0$. Then we look for the upper bound of $\frac{1}{|\mathcal{V}_\mathsf{R}|} \sum_{i \in \mathcal{V}_\mathsf{R}} \mathbb{E}\|x_i^{t+1} - \bar{x}^t\|_2^2$ in terms of $\rho^2$, $p$, and $\delta_{\max}$.

$$\frac{1}{|\mathcal{V}_\mathsf{R}|} \sum_{i \in \mathcal{V}_\mathsf{R}} \mathbb{E}\|x_i^{t+1} - \bar{x}^t\|_2^2 = \frac{1}{|\mathcal{V}_\mathsf{R}|} \sum_{i \in \mathcal{V}_\mathsf{R}} \mathbb{E}\|\sum_{j=1}^n W_{ij} z_{j \to i}^{t+1} - \bar{x}^t\|_2^2$$

$$= \frac{1}{|\mathcal{V}_\mathsf{R}|} \sum_{i \in \mathcal{V}_\mathsf{R}} \mathbb{E}\|(\sum_{j \in \mathcal{V}_\mathsf{R}} \widetilde{W}_{ij} x_j^t - \bar{x}^t) + (\sum_{j=1}^n W_{ij} z_{j \to i}^{t+1} - \sum_{j \in \mathcal{V}_\mathsf{R}} \widetilde{W}_{ij} x_j^t)\|_2^2.$$

Apply (16) with $\varepsilon > 0$ and use the expected improvement Lemma 4

$$\frac{1}{|\mathcal{V}_\mathsf{R}|} \sum_{i \in \mathcal{V}_\mathsf{R}} \mathbb{E}\|x_i^{t+1} - \bar{x}^t\|_2^2$$

$$\leq \frac{1+\varepsilon}{|\mathcal{V}_\mathsf{R}|} \sum_{i \in \mathcal{V}_\mathsf{R}} \|\sum_{j \in \mathcal{V}_\mathsf{R}} \widetilde{W}_{ij} x_j^t - \bar{x}^t\|_2^2 + \frac{1+\frac{1}{\varepsilon}}{|\mathcal{V}_\mathsf{R}|} \sum_{i \in \mathcal{V}_\mathsf{R}} \mathbb{E}\|\sum_{j=1}^n W_{ij} z_{j \to i}^{t+1} - \sum_{j \in \mathcal{V}_\mathsf{R}} \widetilde{W}_{ij} x_j^t\|_2^2$$

$$\leq \frac{(1+\varepsilon)(1-p)}{|\mathcal{V}_\mathsf{R}|} \sum_{i \in \mathcal{V}_\mathsf{R}} \|x_i^t - \bar{x}^t\|_2^2 + \frac{1+\frac{1}{\varepsilon}}{|\mathcal{V}_\mathsf{R}|} \sum_{i \in \mathcal{V}_\mathsf{R}} \mathbb{E}\|\sum_{j=1}^n W_{ij} z_{j \to i}^{t+1} - \sum_{j \in \mathcal{V}_\mathsf{R}} \widetilde{W}_{ij} x_j^t\|_2^2$$

$$\leq (1+\varepsilon)(1-p)\Xi^t + \frac{1+\frac{1}{\varepsilon}}{|\mathcal{V}_\mathsf{R}|} \sum_{i \in \mathcal{V}_\mathsf{R}} \mathbb{E}\|\sum_{j=1}^n W_{ij} z_{j \to i}^{t+1} - \sum_{j \in \mathcal{V}_\mathsf{R}} \widetilde{W}_{ij} x_j^t\|_2^2$$

Replace $x_j^t = x_j^{t+1/2} + \eta m_j^{t+1}$ using (12), then apply (18) and $\eta = 0$

$$\frac{1}{|\mathcal{V}_\mathsf{R}|} \sum_{i \in \mathcal{V}_\mathsf{R}} \mathbb{E}\|x_i^{t+1} - \bar{x}^t\|_2^2 \leq (1+\varepsilon)(1-p)\Xi^t + \frac{1+\frac{1}{\varepsilon}}{|\mathcal{V}_\mathsf{R}|} \sum_{i \in \mathcal{V}_\mathsf{R}} \mathbb{E}\|\sum_{j=1}^n W_{ij} z_{j \to i}^{t+1} - \sum_{j \in \mathcal{V}_\mathsf{R}} \widetilde{W}_{ij} x_j^{t+1/2}\|_2^2.$$

Recall the definition of $e_2^{t+1}$

$$e_2^{t+1} := \frac{1}{|\mathcal{V}_\mathsf{R}|} \sum_{i \in \mathcal{V}_\mathsf{R}} \mathbb{E}\|\sum_{j=1}^n W_{ij} z_{j \to i}^{t+1} - \sum_{j \in \mathcal{V}_\mathsf{R}} \widetilde{W}_{ij} x_j^{t+1/2}\|_2^2.$$

Then use Lemma 9 with the case $A = \widetilde{W}$ and apply Lemma 10 with $\eta = 0$

$$\frac{1}{|\mathcal{V}_\mathsf{R}|} \sum_{i \in \mathcal{V}_\mathsf{R}} \mathbb{E}\|x_i^{t+1} - \bar{x}^t\|_2^2 \leq (1+\varepsilon)(1-p)\Xi^t + (1+\frac{1}{\varepsilon})e_2^{t+1} \leq (1+\varepsilon)(1-p)\Xi^t + (1+\frac{1}{\varepsilon})32\delta_{\max}\Xi^t.$$

Let's minimize the right hand side of the above inequality by taking $\varepsilon$ such that $\varepsilon(1-p) = \frac{32\delta_{\max}}{\varepsilon}$ which leads to $\varepsilon = \sqrt{\frac{32\delta_{\max}}{1-p}}$, then the above inequality becomes

$$\frac{1}{|\mathcal{V}_\mathsf{R}|} \sum_{i \in \mathcal{V}_\mathsf{R}} \mathbb{E}\|x_i^{t+1} - \bar{x}^t\|_2^2 \leq (1 - p + 32\delta_{\max} + 2\sqrt{32\delta_{\max}(1-p)})\Xi^t = (\sqrt{1-p} + \sqrt{32\delta_{\max}})^2 \Xi^t.$$

The consensus distance to the average consensus is only guaranteed to reduce if $\sqrt{1-p} + \sqrt{32\delta_{\max}} < 1$ which is

$$\delta_{\max} < \frac{1}{32}(1 - \sqrt{1-p})^2.$$

Finally, we complete the proof by simplifying the notation to spectral gap $\gamma := 1 - \sqrt{1-p}$. □

Recall that

$$e_2^{t+1} := \frac{1}{|\mathcal{V}_\mathsf{R}|} \sum_{i \in \mathcal{V}_\mathsf{R}} \left\| \sum_{j \in \mathcal{V}_\mathsf{R}} W_{ij}(z_{j \to i}^{t+1} - x_j^{t+1/2}) + \sum_{j \in \mathcal{V}_\mathsf{B}} W_{ij}(z_{j \to i}^{t+1} - x_i^{t+1/2}) \right\|_2^2. \quad (25)$$

Next we consider the bound on consensus distance $\Xi^t$.

**Lemma 11** (Bound consensus distance $\Xi^t$). *Assume Lemma 4, then $\Xi^t$ has the following iteration*

$$\Xi^{t+1} \leq (1+\varepsilon)(1-p)\Xi^t + c_2(1+\frac{1}{\varepsilon})\left(e_2^{t+1} + \eta^2\bar{e}_1^{t+1} + \eta^2\zeta^2 + \eta^2\|\nabla f(\bar{\boldsymbol{x}}^t)\|_2^2 + \eta^2\,\mathbb{E}\|\bar{\boldsymbol{m}}^{t+1} - \frac{1}{\eta}\Delta^{t+1}\|_2^2\right).$$

*where $\varepsilon > 0$ is determined later such that $(1+\varepsilon)(1-p) < 1$ and $c_2 = 5$.*

*Proof.* Expand the consensus distance at time $t+1$

$$\Xi^{t+1} = \frac{1}{|\mathcal{V}_{\mathsf{R}}|}\sum_{i\in\mathcal{V}_{\mathsf{R}}}\mathbb{E}\|\boldsymbol{x}_i^{t+1} - \bar{\boldsymbol{x}}^{t+1}\|_2^2 = \frac{1}{|\mathcal{V}_{\mathsf{R}}|}\sum_{i\in\mathcal{V}_{\mathsf{R}}}\mathbb{E}\|\sum_{j=1}^n \boldsymbol{W}_{ij}\boldsymbol{z}_{j\to i}^{t+1} - \bar{\boldsymbol{x}}^{t+1}\|_2^2$$

$$= \frac{1}{|\mathcal{V}_{\mathsf{R}}|}\sum_{i\in\mathcal{V}_{\mathsf{R}}}\mathbb{E}\|\sum_{j=1}^n \boldsymbol{W}_{ij}\boldsymbol{z}_{j\to i}^{t+1} - \bar{\boldsymbol{x}}^t + \bar{\boldsymbol{x}}^t - \bar{\boldsymbol{x}}^{t+1}\|_2^2$$

$$= \frac{1}{|\mathcal{V}_{\mathsf{R}}|}\sum_{i\in\mathcal{V}_{\mathsf{R}}}\mathbb{E}\|(\sum_{j\in\mathcal{V}_{\mathsf{R}}}\widetilde{\boldsymbol{W}}_{ij}\boldsymbol{x}_j^t - \bar{\boldsymbol{x}}^t) + (\sum_{j=1}^n \boldsymbol{W}_{ij}\boldsymbol{z}_{j\to i}^{t+1} - \sum_{j\in\mathcal{V}_{\mathsf{R}}}\widetilde{\boldsymbol{W}}_{ij}\boldsymbol{x}_j^t) + \bar{\boldsymbol{x}}^t - \bar{\boldsymbol{x}}^{t+1}\|_2^2.$$

Apply Young's inequality (16) with coefficient $\varepsilon$, like the proof of Theorem I, and use the expected improvement Lemma 4

$$\Xi^{t+1} \leq \frac{1+\varepsilon}{|\mathcal{V}_{\mathsf{R}}|}\sum_{i\in\mathcal{V}_{\mathsf{R}}}\|\sum_{j\in\mathcal{V}_{\mathsf{R}}}\widetilde{\boldsymbol{W}}_{ij}\boldsymbol{x}_j^t - \bar{\boldsymbol{x}}^t\|_2^2$$

$$+ \frac{1+\varepsilon}{\varepsilon|\mathcal{V}_{\mathsf{R}}|}\sum_{i\in\mathcal{V}_{\mathsf{R}}}\mathbb{E}\|\sum_{j=1}^n \boldsymbol{W}_{ij}\boldsymbol{z}_{j\to i}^{t+1} - \sum_{j\in\mathcal{V}_{\mathsf{R}}}\widetilde{\boldsymbol{W}}_{ij}\boldsymbol{x}_j^t + \bar{\boldsymbol{x}}^t - \bar{\boldsymbol{x}}^{t+1}\|_2^2$$

$$\leq \frac{(1+\varepsilon)(1-p)}{|\mathcal{V}_{\mathsf{R}}|}\sum_{i\in\mathcal{V}_{\mathsf{R}}}\|\boldsymbol{x}_i^t - \bar{\boldsymbol{x}}^t\|_2^2 + \frac{1+\varepsilon}{\varepsilon|\mathcal{V}_{\mathsf{R}}|}\sum_{i\in\mathcal{V}_{\mathsf{R}}}\mathbb{E}\|\sum_{j=1}^n \boldsymbol{W}_{ij}\boldsymbol{z}_{j\to i}^{t+1} - \sum_{j\in\mathcal{V}_{\mathsf{R}}}\widetilde{\boldsymbol{W}}_{ij}\boldsymbol{x}_j^t + \bar{\boldsymbol{x}}^t - \bar{\boldsymbol{x}}^{t+1}\|_2^2$$

$$\leq (1+\varepsilon)(1-p)\Xi^t + \underbrace{\frac{1+\varepsilon}{\varepsilon|\mathcal{V}_{\mathsf{R}}|}\sum_{i\in\mathcal{V}_{\mathsf{R}}}\mathbb{E}\|(\sum_{j=1}^n \boldsymbol{W}_{ij}\boldsymbol{z}_{j\to i}^{t+1} - \sum_{j\in\mathcal{V}_{\mathsf{R}}}\widetilde{\boldsymbol{W}}_{ij}\boldsymbol{x}_j^t) + \bar{\boldsymbol{x}}^t - \bar{\boldsymbol{x}}^{t+1}\|_2^2}_{=:T_1}$$

Replace $\boldsymbol{x}_j^t = \boldsymbol{x}_j^{t+1/2} + \eta\boldsymbol{m}_j^{t+1}$ using (12), then apply (18)

$$T_1 = \frac{1+\varepsilon}{\varepsilon|\mathcal{V}_{\mathsf{R}}|}\sum_{i\in\mathcal{V}_{\mathsf{R}}}\mathbb{E}\|\sum_{j=1}^n \boldsymbol{W}_{ij}\boldsymbol{z}_{j\to i}^{t+1} - \sum_{j\in\mathcal{V}_{\mathsf{R}}}\widetilde{\boldsymbol{W}}_{ij}\boldsymbol{x}_j^{t+1/2} - \eta\sum_{j\in\mathcal{V}_{\mathsf{R}}}\widetilde{\boldsymbol{W}}_{ij}\boldsymbol{m}_j^{t+1} + \bar{\boldsymbol{x}}^t - \bar{\boldsymbol{x}}^{t+1}\|_2^2$$

$$\leq 5\frac{1+\varepsilon}{\varepsilon}\left(\frac{1}{|\mathcal{V}_{\mathsf{R}}|}\sum_{i\in\mathcal{V}_{\mathsf{R}}}\mathbb{E}\|\sum_{j=1}^n \boldsymbol{W}_{ij}\boldsymbol{z}_{j\to i}^{t+1} - \sum_{j\in\mathcal{V}_{\mathsf{R}}}\widetilde{\boldsymbol{W}}_{ij}\boldsymbol{x}_j^{t+1/2}\|_2^2\right.$$

$$\left.+ \frac{\eta^2}{|\mathcal{V}_{\mathsf{R}}|}\sum_{i\in\mathcal{V}_{\mathsf{R}}}\mathbb{E}\|\sum_{j\in\mathcal{V}_{\mathsf{R}}}\widetilde{\boldsymbol{W}}_{ij}(\boldsymbol{m}_j^{t+1} - \nabla f_j(\bar{\boldsymbol{x}}^t))\|_2^2\right. \tag{26}$$

$$\left.+ \frac{\eta^2}{|\mathcal{V}_{\mathsf{R}}|}\sum_{i\in\mathcal{V}_{\mathsf{R}}}\|\sum_{j\in\mathcal{V}_{\mathsf{R}}}\widetilde{\boldsymbol{W}}_{ij}\nabla f_j(\bar{\boldsymbol{x}}^t) - \nabla f(\bar{\boldsymbol{x}}^t)\|_2^2 + \eta^2\|\nabla f(\bar{\boldsymbol{x}}^t)\|_2^2 + \mathbb{E}\|\bar{\boldsymbol{x}}^t - \bar{\boldsymbol{x}}^{t+1}\|_2^2\right).$$

Recall the definition of $e_2^{t+1}$

$$e_2^{t+1} := \frac{1}{|\mathcal{V}_{\mathsf{R}}|}\sum_{i\in\mathcal{V}_{\mathsf{R}}}\mathbb{E}\left\|\sum_{j\in\mathcal{V}_{\mathsf{R}}}\boldsymbol{W}_{ij}(\boldsymbol{z}_{j\to i}^{t+1} - \boldsymbol{x}_j^{t+1/2}) + \sum_{j\in\mathcal{V}_{\mathsf{B}}}\boldsymbol{W}_{ij}(\boldsymbol{z}_{j\to i}^{t+1} - \boldsymbol{x}_i^{t+1/2})\right\|_2^2$$

$$= \frac{1}{|\mathcal{V}_{\mathsf{R}}|}\sum_{i\in\mathcal{V}_{\mathsf{R}}}\mathbb{E}\|\sum_{j=1}^n \boldsymbol{W}_{ij}\boldsymbol{z}_{j\to i}^{t+1} - \sum_{j\in\mathcal{V}_{\mathsf{R}}}\widetilde{\boldsymbol{W}}_{ij}\boldsymbol{x}_j^{t+1/2}\|_2^2$$

Then use Lemma 9 with the case $A = \widetilde{W}$,

$$T_1 \leq 5(1 + \frac{1}{\varepsilon}) \left( e_2^{t+1} + \eta^2 \bar{e}_1^{t+1} + \frac{\eta^2}{|\mathcal{V}_\mathsf{R}|} \sum_{i \in \mathcal{V}_\mathsf{R}} \| \sum_{j \in \mathcal{V}_\mathsf{R}} \widetilde{W}_{ij} \nabla f_j(\bar{x}^t) - \nabla f(\bar{x}^t) \|_2^2 + \eta^2 \| \nabla f(\bar{x}^t) \|_2^2 + \mathbb{E} \| \bar{x}^t - \bar{x}^{t+1} \|_2^2 \right).$$

Use convexity of $\|\cdot\|_2^2$ and (A3) we have

$$T_1 \leq 5(1 + \frac{1}{\varepsilon}) \left( e_2^{t+1} + \eta^2 \bar{e}_1^{t+1} + \eta^2 \zeta^2 + \eta^2 \| \nabla f(\bar{x}^t) \|_2^2 + \mathbb{E} \| \bar{x}^t - \bar{x}^{t+1} \|_2^2 \right).$$

Use (21) for the last term

$$T_1 \leq 5(1 + \frac{1}{\varepsilon}) \left( e_2^{t+1} + \eta^2 \bar{e}_1^{t+1} + \eta^2 \zeta^2 + \eta^2 \| \nabla f(\bar{x}^t) \|_2^2 + \eta^2 \, \mathbb{E} \| \bar{m}^{t+1} - \frac{1}{\eta} \Delta^{t+1} \|_2^2 \right).$$

Finally, by the definition of $\tilde{e}_1^{t+1}$, we have

$$\Xi^{t+1} \leq (1 + \varepsilon)(1 - p) \Xi^t + 5(1 + \frac{1}{\varepsilon}) \left( e_2^{t+1} + \eta^2 \bar{e}_1^{t+1} + \eta^2 \zeta^2 + \eta^2 \| \nabla f(\bar{x}^t) \|_2^2 + \eta^2 \, \mathbb{E} \| \bar{m}^{t+1} - \frac{1}{\eta} \Delta^{t+1} \|_2^2 \right).$$

$\square$

**Lemma 12** (Tuning stepsize.). *Suppose the following holds for any step size $\eta \leq d$:*

$$\Psi_T \leq \frac{r_0}{\eta(T+1)} + b\eta + e\eta^2 + f\eta^3.$$

*Then, there exists a step-size $\eta \leq d$ such that*

$$\Psi_T \leq 2 \left( \frac{br_0}{T+1} \right)^{\frac{1}{2}} + 2e^{\frac{1}{3}} \left( \frac{r_0}{T+1} \right)^{\frac{2}{3}} + 2f^{\frac{1}{4}} \left( \frac{r_0}{T+1} \right)^{\frac{3}{4}} + \frac{dr_0}{T+1}.$$

*Proof.* Choosing $\eta = \min \left\{ \left( \frac{r_0}{b(T+1)} \right)^{\frac{1}{2}}, \left( \frac{r_0}{e(T+1)} \right)^{\frac{1}{3}}, \left( \frac{r_0}{f(T+1)} \right)^{\frac{1}{4}}, \frac{1}{d} \right\} \leq \frac{1}{d}$ we have four cases

- $\eta = \frac{1}{d}$ and is smaller than $\left( \frac{r_0}{b(T+1)} \right)^{\frac{1}{2}}, \left( \frac{r_0}{e(T+1)} \right)^{\frac{1}{3}}, \left( \frac{r_0}{f(T+1)} \right)^{\frac{1}{4}}$, then

$$\Psi_T \leq \frac{dr_0}{T+1} + \frac{b}{d} + \frac{e}{d^2} + \frac{f}{d^3} \leq \frac{dr_0}{T+1} + \left( \frac{br_0}{T+1} \right)^{\frac{1}{2}} + e^{1/3} \left( \frac{r_0}{T+1} \right)^{\frac{2}{3}} + f^{1/4} \left( \frac{r_0}{T+1} \right)^{\frac{3}{4}}.$$

- $\eta = \left( \frac{r_0}{b(T+1)} \right)^{\frac{1}{2}} < \min\{ \left( \frac{r_0}{e(T+1)} \right)^{\frac{1}{3}}, \left( \frac{r_0}{f(T+1)} \right)^{\frac{1}{4}} \}$, then

$$\Psi_T \leq 2 \left( \frac{br_0}{T+1} \right)^{\frac{1}{2}} + \frac{er_0}{b(T+1)} + f \left( \frac{r_0}{b(T+1)} \right)^{\frac{3}{2}} \leq 2 \left( \frac{br_0}{bT+1} \right)^{\frac{1}{2}} + e^{1/3} \left( \frac{r_0}{T+1} \right)^{\frac{2}{3}} + f^{1/4} \left( \frac{r_0}{T+1} \right)^{\frac{3}{4}}.$$

- $\eta = \left( \frac{r_0}{e(T+1)} \right)^{\frac{1}{3}} < \min\{ \left( \frac{r_0}{b(T+1)} \right)^{\frac{1}{2}}, \left( \frac{r_0}{f(T+1)} \right)^{\frac{1}{4}} \}$, then

$$\Psi_T \leq 2e^{1/3} \left( \frac{r_0}{T+1} \right)^{\frac{2}{3}} + b \left( \frac{r_0}{e(T+1)} \right)^{\frac{1}{3}} + \frac{fr_0}{e(T+1)} \leq \left( \frac{br_0}{T+1} \right)^{\frac{1}{2}} + 2e^{1/3} \left( \frac{r_0}{T+1} \right)^{\frac{2}{3}} + f^{1/4} \left( \frac{r_0}{T+1} \right)^{\frac{3}{4}}.$$

- $\eta = \left( \frac{r_0}{f(T+1)} \right)^{\frac{1}{4}} < \min\{ \left( \frac{r_0}{b(T+1)} \right)^{\frac{1}{2}}, \left( \frac{r_0}{e(T+1)} \right)^{\frac{1}{3}} \}$, then

$$\Psi_T \leq 2f^{1/4} \left( \frac{r_0}{T+1} \right)^{\frac{3}{4}} + b \left( \frac{r_0}{f(T+1)} \right)^{\frac{1}{4}} + e \left( \frac{r_0}{f(T+1)} \right)^{\frac{1}{2}} \leq \left( \frac{br_0}{T+1} \right)^{\frac{1}{2}} + e^{1/3} \left( \frac{r_0}{T+1} \right)^{\frac{2}{3}} + 2f^{1/4} \left( \frac{r_0}{T+1} \right)^{\frac{3}{4}}.$$

Then, take the uniform upper bound of the upper bound gives the result. $\square$

### E.3 Proof of the main theorem

**Theorem III′.** *Suppose Assumptions 1–4 hold and $\delta_{\max} = \mathcal{O}(\gamma^2)$. Define the clipping radius as*

$$\tau_i^{t+1} = \sqrt{\frac{1}{\delta_i} \sum_{j \in \mathcal{V}_\mathsf{R}} \boldsymbol{W}_{ij} \, \mathbb{E} \left\| \boldsymbol{x}_i^{t+1/2} - \boldsymbol{x}_j^{t+1/2} \right\|_2^2}. \tag{27}$$

*Then for $\alpha := 3\eta L$, the iterates of Algorithm 1 satisfy*

$$\frac{1}{T+1} \sum_{t=0}^{T} \|\nabla f(\bar{\boldsymbol{x}}^t)\|_2^2 \leq \frac{200 c_1 c_2}{\gamma^2} \delta_{\max} \zeta^2 + 2 \left( \frac{3^2}{|\mathcal{V}_\mathsf{R}|} + \frac{320 c_1 c_2}{\gamma^2} \delta_{\max} \right)^{1/2} \left( \frac{3 L \sigma^2 r_0}{T+1} \right)^{1/2}$$

$$+ 2 \left( \frac{48 c_2}{\gamma^2} \zeta^2 \right)^{1/3} \left( \frac{r_0 L}{T+1} \right)^{2/3} + 2 \left( \frac{144 c_2}{\gamma^2} \sigma^2 \right)^{1/4} \left( \frac{r_0 L}{T+1} \right)^{3/4} + \frac{d_0 r_0}{T+1}.$$

*where $r_0 := f(\boldsymbol{x}^0) - f^\star$ and $c_1 = 32$ and $c_2 = 5$. Furthermore, the consensus distance has an upper bound*

$$\frac{1}{|\mathcal{V}_\mathsf{R}|} \sum_{i \in \mathcal{V}_\mathsf{R}} \|\boldsymbol{x}_i^t - \bar{\boldsymbol{x}}^t\|_2^2 = \mathcal{O}\left( \frac{\zeta^2}{\gamma^2 (T+1)} \right).$$

**Remark 13.** *The requirement $\delta_{\max} = \mathcal{O}(\gamma^2)$ suggest that $\delta_{\max}$ and $\gamma^2$ are of same order. The exact constant are determined in the proof and can be tighten simply through better constants in equalities like (23), (26). In practice* CLIPPEDGOSSIP *allow high number of attackers. For example in Figure 15, 1/6 of workers are Byzantine and* CLIPPEDGOSSIP *still perform well in the non-IID setting.*

*Proof.* Denote the terms of average $t$ from $0$ to $T$ as follows

$$C_1 := \frac{1}{1+T} \sum_{t=0}^{T} \|\nabla f(\bar{\boldsymbol{x}}^t)\|_2^2, \, C_2 := \frac{1}{1+T} \sum_{t=0}^{T} \|\bar{\boldsymbol{m}}^{t+1} - \frac{1}{\eta} \Delta^{t+1}\|_2^2, \, D_1 := \frac{1}{1+T} \sum_{t=0}^{T} \Xi^{t+1}$$

$$E_1 := \frac{1}{1+T} \sum_{t=0}^{T} e_1^{t+1}, \, \bar{E}_1 := \frac{1}{1+T} \sum_{t=0}^{T} \bar{e}_1^{t+1}, \, E_I := \frac{1}{1+T} \sum_{t=0}^{T} e_I^{t+1}, \, E_2 := \frac{1}{1+T} \sum_{t=0}^{T} e_2^{t+1}$$

First we apply average to Lemma 10

$$E_2 \leq c_2 \delta_{\max} \left( 2\eta^2 (E_I + \zeta^2) + D_1 \right). \tag{28}$$

Then we rewrite key Lemma 8 as

$$\|\nabla f(\bar{\boldsymbol{x}}^t)\|_2^2 + \frac{1}{2} \mathbb{E} \|\bar{\boldsymbol{m}}^{t+1} - \frac{1}{\eta} \Delta^{t+1}\|_2^2 \leq \frac{2}{\eta} (r^t - r^{t+1}) + 2 e_1^{t+1} + \frac{2}{\eta^2} e_2^{t+1},$$

and further average over time $t$

$$C_1 + \frac{1}{2} C_2 \leq \frac{2 r_0}{\eta (T+1)} + 2 E_1 + \frac{2}{\eta^2} E_2$$

where we use $-f(\boldsymbol{x}^{T+1}) \leq -f^\star$. Combined with (28) gives

$$C_1 + \frac{1}{2} C_2 \leq \frac{2 r_0}{\eta (T+1)} + 2 E_1 + 4 c_2 \delta_{\max} E_I + 4 c_2 \delta_{\max} \zeta^2 + \frac{2 c_2 \delta_{\max}}{\eta^2} D_1 \tag{29}$$

Now we also average Lemma 9 for $e_1^{t+1}$ over $t$ gives

$$\frac{1}{1+T} \sum_{t=0}^{T} e_1^{t+1} \leq \frac{1-\alpha}{1+T} \sum_{t=0}^{T} e_1^t + 2\alpha L^2 D_1 + \frac{\alpha^2 \sigma^2}{|\mathcal{V}_\mathsf{R}|} + \frac{2 L^2 \eta^2}{\alpha} \frac{1}{1+T} \sum_{t=0}^{T} \|\bar{\boldsymbol{m}}^t - \frac{1}{\eta} \Delta^t\|_2^2$$

$$\leq \frac{1-\alpha}{1+T} \sum_{t=0}^{T} e_1^{t+1} + 2\alpha L^2 D_1 + \frac{\alpha^2 \sigma^2}{|\mathcal{V}_\mathsf{R}|} + \frac{2 L^2 \eta^2}{\alpha} C_2$$

where we use $\Xi^0 = e_1^0 = 0$ and $\bar{\boldsymbol{m}}^0 = \Delta^0 = \boldsymbol{0}$. Then let $\beta_1 := \frac{2 L^2 \eta^2}{\alpha^2}$

$$E_1 \leq 2 L^2 D_1 + \frac{\alpha \sigma^2}{|\mathcal{V}_\mathsf{R}|} + \beta_1 C_2. \tag{30}$$

Similarly, Lemma 9 for $e_{\boldsymbol{I}}^{t+1}$ the only difference is that we don't have $\frac{1}{n}$ for $\sigma^2$

$$E_{\boldsymbol{I}} \leq 2L^2 D_1 + \alpha\sigma^2 + \beta_1 C_2. \tag{31}$$

Similarly, let's call $\beta_2 := \frac{1}{|\mathcal{V}_R|}\sum_{i\in\mathcal{V}_R}\sum_{j\in\mathcal{V}_R}\widetilde{\boldsymbol{W}}_{ij}^2 \leq 1$

$$\bar{E}_1 \leq 2L^2 D_1 + \beta_2\alpha\sigma^2 + \beta_1 C_2. \tag{32}$$

The consensus distance Lemma 11 has

$$D_1 \leq \frac{(1+\varepsilon)(1-p)}{1+T}\sum_{t=0}^{T}\Xi^t + c_2(1+\tfrac{1}{\varepsilon})E_2 + c_2(1+\tfrac{1}{\varepsilon})\eta^2(\bar{E}_1^{t+1} + \zeta^2 + C_1 + C_2)$$

$$\leq (1+\varepsilon)(1-p)D_1 + c_2(1+\tfrac{1}{\varepsilon})E_2 + c_2(1+\tfrac{1}{\varepsilon})\eta^2(\bar{E}_1^{t+1} + \zeta^2 + C_1 + C_2).$$

Replace $E_2$ using (28) gives

$$D_1 \leq (1+\varepsilon)(1-p)D_1 + c_2(1+\tfrac{1}{\varepsilon})(c_1\delta_{\max}(2\eta^2(E_{\boldsymbol{I}}^{t+1} + \zeta^2) + D_1)) + c_2(1+\tfrac{1}{\varepsilon})\eta^2(\bar{E}_1^{t+1} + \zeta^2 + C_1 + C_2)$$

$$\leq ((1+\varepsilon)(1-p) + c_1c_2(1+\tfrac{1}{\varepsilon})\delta_{\max})D_1 + c_2(1+\tfrac{1}{\varepsilon})\eta^2(2c_1\delta_{\max}E_{\boldsymbol{I}}^{t+1} + \bar{E}_1^{t+1} + (1+2c_1\delta_{\max})\zeta^2 + C_1 + C_2).$$

Now replace $\bar{E}_1$, $E_{\boldsymbol{I}}$ with (32), (31), then

$$D_1 \leq ((1+\varepsilon)(1-p) + c_2(1+\tfrac{1}{\varepsilon})(c_1\delta_{\max}(1+4L^2\eta^2) + 2L^2\eta^2))D_1$$

$$+ c_2(1+\tfrac{1}{\varepsilon})\eta^2((2c_1\delta_{\max} + \beta_2)\alpha\sigma^2 + (2c_1\delta_{\max} + 1)\zeta^2 + ((2c_1\delta_{\max} + 1)\beta_1 + 1)C_2 + C_1).$$

By enforcing $\eta \leq \frac{\gamma}{9L}$ and $\delta_{\max} \leq \frac{\gamma^2}{10c_1c_2}$ we have

$$2c_2L^2\eta^2 \leq \gamma^2/8$$

$$c_1c_2\delta_{\max}(1+4L^2\eta^2) \leq \gamma^2/8$$

we can achieve

$$\sqrt{c_1c_2\delta_{\max}(1+4L^2\eta^2) + 2c_2L^2\eta^2} \leq \frac{\gamma}{2}.$$

Then

$$D_1 \leq \underbrace{\left((1+\varepsilon)(1-p) + (1+\tfrac{1}{\varepsilon})\tfrac{\gamma^2}{4}\right)}_{=:T_2}D_1$$

$$+ c_2(1+\tfrac{1}{\varepsilon})\eta^2((2c_1\delta_{\max} + \beta_2)\alpha\sigma^2 + (2c_1\delta_{\max} + 1)\zeta^2 + ((2c_1\delta_{\max} + 1)\beta_1 + 1)C_2 + C_1).$$

Let us minimize the the coefficients of $D_1$ on the right hand side of inequality by having

$$\varepsilon(1-p) = \frac{1}{\varepsilon}\frac{\gamma^2}{4},$$

that is $\varepsilon = \sqrt{\frac{\gamma^2}{4(1-p)}}$. Then the coefficient becomes

$$T_2 = (1+\varepsilon)(1-p) + (1+\tfrac{1}{\varepsilon})\frac{\gamma^2}{4}$$

$$= (\sqrt{1-p} + \frac{\gamma}{2})^2$$

$$= (1 - \frac{\gamma}{2})^2.$$

Then we use $\frac{1}{\varepsilon} = \sqrt{\frac{4(1-p)}{\gamma^2}} \leq \frac{2}{\gamma}$ and $1 + \frac{1}{\varepsilon} \leq \frac{3}{\gamma}$

$$D_1 \leq \frac{4c_2\eta^2}{\gamma^2}((2c_1\delta_{\max} + \beta_2)\alpha\sigma^2 + (2c_1\delta_{\max} + 1)\zeta^2 + ((2c_1\delta_{\max} + 1)\beta_1 + 1)C_2 + C_1).$$

This leads to $2c_1\delta_{\max} \leq \frac{\gamma^2}{5c_2} \leq 1$ and $\beta_2 \leq 1$, then we know

$$D_1 \leq \frac{4c_2\eta^2}{\gamma^2}(2\alpha\sigma^2 + 2\zeta^2 + C_1 + (1+2\beta_1)C_2) \tag{33}$$

Finally, we combine (29), (30), (32)

$$C_1 + \frac{1}{2}C_2 \leq \frac{2r_0}{\eta(T+1)} + 2E_1 + 4c_1\delta_{\max}E_I + 4c_1\delta_{\max}\zeta^2 + \frac{2c_1\delta_{\max}}{\eta^2}D_1$$

$$\leq \frac{2r_0}{\eta(T+1)} + (4L^2D_1 + \frac{2\alpha\sigma^2}{|\mathcal{V}_R|} + 2\beta_1 C_2) + 2c_1\delta_{\max}(4L^2 D_1 + 2\beta_2\alpha\sigma^2 + 2\beta_1 C_2)$$

$$+ 4c_1\delta_{\max}\zeta^2 + \frac{2c_1\delta_{\max}}{\eta^2}D_1$$

$$\leq \frac{2r_0}{\eta(T+1)} + (4L^2 + 8c_1\delta_{\max}L^2 + \frac{2c_1\delta_{\max}}{\eta^2})D_1 + (\frac{1}{|\mathcal{V}_R|} + 2c_1\delta_{\max})2\alpha\sigma^2$$

$$+ 4\beta_1 C_2 + 4c_1\delta_{\max}\zeta^2$$

Then we replace $D_1$ with (33)

$$\begin{aligned} C_1 + \frac{1}{2}C_2 \leq & \frac{2r_0}{\eta(T+1)} + (\frac{1}{|\mathcal{V}_R|} + 2c_1\delta_{\max})2\alpha\sigma^2 + 4\beta_1 C_2 + 4c_1\delta_{\max}\zeta^2 \\ & + (4L^2\eta^2 + 8c_1\delta_{\max}L^2\eta^2 + 2c_1\delta_{\max})\frac{4c_2}{\gamma^2}(2\alpha\sigma^2 + 2\zeta^2 + C_1 + (1+2\beta_1)C_2) \end{aligned} \tag{34}$$

To have a valid bound on $C_1$, there are two constraints on the coefficient of the RHS $C_1$ and $C_2$.

$$(4L^2\eta^2 + 8c_1\delta_{\max}L^2\eta^2 + 2c_1\delta_{\max})\frac{4c_2}{\gamma^2} < 1$$

$$(4L^2\eta^2 + 8c_1\delta_{\max}L^2\eta^2 + 2c_1\delta_{\max})\frac{4c_2}{\gamma^2}(1+2\beta_1) + 4\beta_1 \leq \frac{1}{2}.$$

We can strength the first requirement to

$$(4L^2\eta^2 + 8c_1\delta_{\max}L^2\eta^2 + 2c_1\delta_{\max})\frac{4c_2}{\gamma^2} \leq \frac{1}{4}. \tag{35}$$

Then, apply this inequality to the second inequality gives

$$\frac{1}{4} + \frac{1}{2}\beta_1 + 4\beta_1 \leq \frac{1}{2}$$

which requires $\eta \leq \frac{\alpha}{3L}$. Next (35) can be achieved by requiring $\delta_{\max} \leq \frac{\gamma^2}{64c_1c_2}$

$$(4 + 8c_1\delta_{\max})L^2\eta^2 + 2c_1\delta_{\max} \leq 8L^2\eta^2 + 2c_1\delta_{\max} \leq \frac{\gamma^2}{16c_2}$$

which requires $8\eta^2 L^2 \leq \frac{\gamma^2}{32c_2}$, and we can simplify it to $\eta \leq \frac{\gamma}{40L}$. Now we can simplify (34) with (35)

$$\begin{aligned} \frac{3}{4}C_1 \leq & \frac{2r_0}{\eta(T+1)} + (\frac{1}{|\mathcal{V}_R|} + 2c_1\delta_{\max})2\alpha\sigma^2 + 4c_1\delta_{\max}\zeta^2 \\ & + (4L^2\eta^2 + 8c_1\delta_{\max}L^2\eta^2 + 2c_1\delta_{\max})\frac{4c_2}{\gamma^2}(2\alpha\sigma^2 + 2\zeta^2) \end{aligned}$$

Multiply both sides with $\frac{4}{3}$ and relax constant $\frac{4}{3}\cdot 2 \leq 3$. Then by taking $\eta \leq \frac{1}{2L}$ we have that

$$C_1 \leq \frac{3r_0}{\eta(T+1)} + (\frac{1}{|\mathcal{V}_R|} + \frac{151}{\gamma^2}2c_1\delta_{\max})3\alpha\sigma^2 + \frac{200c_1c_2}{\gamma^2}\delta_{\max}\zeta^2 + \frac{48c_2}{\gamma^2}(\alpha\sigma^2 + \zeta^2)L^2\eta^2$$

By taking $\alpha := 3\eta L$ and relax the constants we have

$$C_1 \leq \frac{3r_0}{\eta(T+1)} + (\frac{3^2}{|\mathcal{V}_R|} + \frac{320c_1}{\gamma^2}\delta_{\max})L\sigma^2\eta + \frac{48c_2}{\gamma^2}(\alpha\sigma^2 + \zeta^2)L^2\eta^2 + \frac{200c_1c_2}{\gamma^2}\delta_{\max}\zeta^2.$$

Minimize the the right hand side by tuning step size Lemma 12 we have

$$\begin{aligned} \frac{1}{T+1}\sum_{t=0}^{T}\|\nabla f(\bar{\boldsymbol{x}}^t)\|_2^2 \leq & \frac{200c_1c_2}{\gamma^2}\delta_{\max}\zeta^2 + 2\left(\frac{(\frac{3^2}{|\mathcal{V}_R|} + \frac{320c_1}{\gamma^2}\delta_{\max})3L\sigma^2 r_0}{T+1}\right)^{\frac{1}{2}} \\ & + 2\left(\frac{48c_2}{\gamma^2}\zeta^2\right)^{\frac{1}{3}}\left(\frac{r_0 L}{T+1}\right)^{\frac{2}{3}} + 2\left(\frac{144c_2}{\gamma^2}\sigma^2\right)^{\frac{1}{4}}\left(\frac{r_0 L}{T+1}\right)^{\frac{3}{4}} + \frac{d_0 r_0}{T+1} \end{aligned}$$

where $\frac{1}{d_0} := \min\{\frac{1}{2L}, \frac{\gamma}{9L}, \frac{\gamma}{40L}\} = \frac{\gamma}{40L}$ and

$$\eta = \min\left\{\left(\frac{2r_0}{(\frac{9}{|\mathcal{V}_R|} + \frac{320c_1}{\gamma^2}\delta_{\max})L\sigma^2(T+1)}\right)^{1/2}, \left(\frac{2r_0\gamma^2}{48c_2\zeta^2 L^2(T+1)}\right)^{1/3}, \left(\frac{2r_0\gamma^2}{L^3\sigma^2(T+1)}\right)^{1/4}, \frac{1}{d_0}\right\}.$$

**Bound on the consensus distance** $D_1$. Since $\beta_1 = \frac{2L^2\eta^2}{\alpha^2} = \frac{2}{9}$, we can relax (33) to

$$D_1 \leq \frac{4c_2\eta^2}{\gamma^2}(2\alpha\sigma^2 + 2\zeta^2 + 2(1+2\beta_1)(C_1 + \tfrac{1}{2}C_2))$$
$$\leq \frac{4c_2\eta^2}{\gamma^2}(2\alpha\sigma^2 + 2\zeta^2 + 3(C_1 + \tfrac{1}{2}C_2)).$$

For significantly large $T$, we know that $\eta = \alpha = \mathcal{O}(\frac{1}{\sqrt{T+1}})$ and find the upper bound of $2\alpha\sigma^2 + 2\zeta^2 + C_1 + \frac{1}{2}C_2$ with $\mathcal{O}(\zeta^2)$ where higher order terms of $1/T$ are dropped. Therefore, the upper bound on the consensus distance $D_1$ is $\mathcal{O}\left(\frac{\zeta^2}{\gamma^2(T+1)}\right)$. $\qquad\square$

## F    OTHER RELATED WORKS AND DISCUSSIONS

In this section, we add more related works and discussions.

**Byzantine resilient learning with constraints**    Byzantine-robustness is challenging when the training is combined with other constraints, such as asynchrony (Damaskinos et al., 2018; Xie et al., 2020b; Yang & Li, 2021), data heterogeneity (Karimireddy et al., 2021b; Peng & Ling, 2020; Li et al., 2019; Data & Diggavi, 2021), privacy (He et al., 2020; Burkhalter et al., 2021). These works all assume the existence of a central server which can communicate with all regular workers. In this paper, we consider the decentralized setting and focus on the constraint that not all regular workers can communicate with each other.

**More works on decentralized learning.**    Many works focus on compression-techniques (Koloskova et al., 2019; 2020a; Vogels et al., 2020), data heterogeneity (Tang et al., 2018; Vogels et al., 2021; Koloskova et al., 2021), and communication topology (Assran et al., 2019; Ying et al., 2021a).

**Detailed comparison with one line of work.**    Among all the works on robust decentralized training, Sundaram et al. Sundaram & Gharesifard (2018) and Su et al. Su & Vaidya (2016a) and their followup works Yang & Bajwa (2019b;a) have the most similar setup with ours. They are all using the trimmed mean as the aggregator assumptions on the graph. We illustrate our advantages over these methods as follows

1. Their methods (TM) make unrealistic assumptions about the graph while our method is much more relaxed. Their main assumption on the graph has 2 parts: 1) each good node should have at least $2b+1$ neighbors where $b$ is the maximum number of Byzantine workers in the *whole* network; 2) by removing any $b$ edges the good nodes should be connected. This assumption essentially requires the good workers have honest majority *everywhere* and additionally they have to be well connected. This can be hardly enforced in the decentralized environment. In contrast, our method has a weaker condition relating the spectral gap and $\delta$. Our method also works without a honest majority Figure 12. The second part of their assumption exclude common topologies like Dumbbell.

2. TM **fails** to reach consensus even in some **Byzantine-free** graphs (e.g. Dumbbell) while SSClip converges as fast as gossip. For example, TM fails to reach consensus in NonIID setting for MNIST dataset (Figure 4) and even fails in IID setting for CIFAR-10 dataset (Figure 14).

3. We have a clear convergence rate for SGD while they only show asymptotic convergence for GD. In fact, we even improve the state-of-art decentralized SGD analysis (Koloskova et al., 2020b).

4. Our work reveals how the quantitative relation between percentage of Byzantine workers ($\delta$) and information bottleneck ($\gamma$) influence the consensus (see Figure 3 and Theorem I).

5. We propose a novel dissensus attacks that utilize topology information.

6. Impossibility results. Sundaram et al. Sundaram & Gharesifard (2018) and Su et al. Su & Vaidya (2016a) give impossibility results in terms of number of nodes while we give a novel results in terms of spectral gap ($\gamma$).

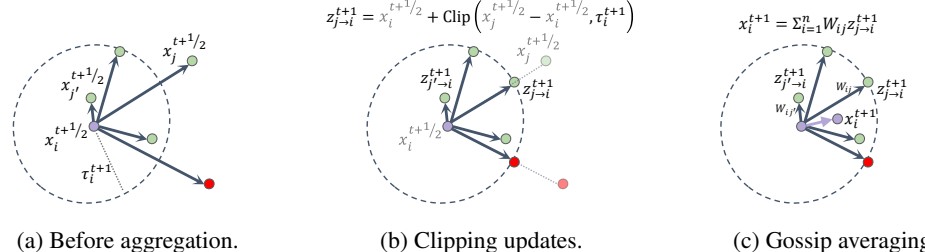

(a) Before aggregation.    (b) Clipping updates.    (c) Gossip averaging.

Figure 17: Diagram of ClippedGossip at time $t$ on worker $i$. Let purple node be the model of worker $i$ and green nodes be models of worker $i$'s regular neighbors and red nodes be models of worker $i$'s Byzantine neighbors. The figure (a), (b), and (c) demonstrate the 3 stages of ClippedGossip. First, in the left figure (a) worker $i$ collects models $\{\boldsymbol{x}_j^{t+1/2} : j \in \mathcal{N}_i\}$ from its neighbors. Then in the middle figure (b) worker $i$ clips neighbor models to ensure the clipped models are no farther than $\tau_i^{t+1}$ from node $i$. Nodes outside the circle (e.g. $\boldsymbol{x}_j^{t+1/2}$) clipped to the circle (e.g. $\boldsymbol{z}_{j\to i}^{t+1}$) while nodes inside the circle (e.g. $\boldsymbol{x}_{j'}^{t+1}$) remain the same after clipping (e.g. $\boldsymbol{z}_{j'\to i}^{t+1}$). In the right figure (c) worker $i$ update its model to $\boldsymbol{x}_i^{t+1}$ using gossip averaging over clipped models.

**Other related works and discussions.**  Zhao et al. Zhao et al. (2019) make assumption that some users are *trusted* and then adopt trimmed mean as robust aggregator. But this assumption is incompatible with our setting where every node only trusts itself. Peng et al. Peng & Ling (2020) propose a "zero-sum" attack which exploits the topology where Byzantine worker $j$ construct

$$\boldsymbol{x}_j := -\frac{\sum_{k \in \mathcal{N}_i \cap \mathcal{V}_R} \boldsymbol{x}_k}{|\mathcal{N}_i \cap \mathcal{V}_B|}.$$

They aim to manipulate the good worker $i$'s model to 0, but it also makes the constructed Byzantine model very far away from the good worker models, making it easy to detect. In contrast, our dissensus attack (6) simply amplifies the existing disagreement amongst the good workers, which keeps the attack much less undetectable. In addition, we take mixing matrix into consideration and use $\varepsilon_i$ to parameterize the attack which makes it more flexible.

**Clarifications about our method.**  We make the following clarifications regarding our method:

- Ideally we would like to replace the $\delta_{\max} = \max_j \delta_j$ with an average $\bar{\delta} = \frac{1}{n}\sum_j \delta_j$. However, the requirement that $\delta_{\max}$ be small may be achieved by the good workers increasing its weight on itself. Note that Byzantine workers cannot alter good workers local behavior.

- Theorem III does not tell us what happens if the percentage of Byzantine workers $\delta$ is relatively larger than spectral gap ($\gamma$), but it does not necessarily mean that CLIPPEDGOSSIP diverges. Instead, it means reaching global consensus is not possible as Byzantine workers effectively block the information bottleneck. We conjecture that within each connected good component not blocked by the byzantine workers, the good workers still reach component-level consensus by applying the analysis of Theorem III to only this component. We leave such a component-wise analysis for future work.

