# OpenReview forum: "Byzantine-robust Decentralized Learning via ClippedGossip"
_ICLR.cc/2023/Conference — Submitted to ICLR 2023_

### Official Review · Reviewer_wSS2 · 2022-10-24

**Confidence:** 4
**Correctness:** 4
**Technical Novelty And Significance:** 2
**Empirical Novelty And Significance:** 2
**Recommendation:** 6

**Clarity, Quality, Novelty And Reproducibility:**

The only problem with this paper is contributions are only marginally significant, and the assistance of this paper in comparison to [1] is not substantial.
[1]Karimireddy, Sai Praneeth, Lie He, and Martin Jaggi. "Byzantine-Robust Learning on Heterogeneous Datasets via Bucketing." arXiv preprint arXiv:2006.09365 (2020).

**Strength And Weaknesses:**

-Weaknesses
1)The aggregation strategy proposed in the paper has been proposed before and is not a novelty of this paper. Moreover, the idea of centralized clipping is not new, for example, paper [1] looked at this idea before in distributed setting, and this paper is an extension of that to the decentralized setting.
[1]Karimireddy, Sai Praneeth, Lie He, and Martin Jaggi. "Byzantine-Robust Learning on Heterogeneous Datasets via Bucketing." arXiv preprint arXiv:2006.09365 (2020).
-Strength
1)The reviewer did a good job reviewing relevant papers.
2)The paper does a good job of motivating the problem.
3)The experimental results are sound.


**Summary Of The Paper:**

This paper studies byzantine robust decentralized learning. In this setting, a fraction of workers is byzantine. First, the paper argues why the consensus is vulnerable to byzantine attacks, then they propose DISSENSUS, a decentralized attack to steer away from the true consensus. Finally, they come up with a robust called aggregation self-centered clipping which converges to a neighborhood of a stationary point.

**Summary Of The Review:**

The aggregation strategy proposed in the paper has been suggested before and is not a novelty of this paper. But, the authors provided a study of this method in a decentralized setting and did a good job comparing it with previous methods and providing experiments that support their idea. Compared to the previous versions of the paper, the authors add more comparisons with previous results in control/optimization literature. In summary, the strengths outweigh the weaknesses.

---

> ### Author Response · Authors · 2022-11-17
> **Author's response to Reviewer wSS2**
>
> We greatly appreciate Reviewer wSS2 for reading through our paper and highlighting the changes we have made since our previous version. We would like to address the novelty issue raised by reviewer wSS2.
>
> We agree with Reviewer wSS2 that the clipping technique has been proposed before. In fact, the clipping technique has been used for different purposes, including Byzantine-robustness (Karimireddy et al., 2020), differential privacy (Abadi et al., 2016; Chen et al., 2020), accelerating training (Zhang et al., 2019), etc. **We would like to clarify that our goal is not to find a new aggregator but rather to find an aggregator which is provably robust in the decentralized optimization.** We show that clipping is robust in the decentralized optimization and enjoys a state-of-the-art convergence rate.
>
> Despite using results from the easier federated case (Karimireddy et al., 2020) in terms of the aggregator, our work studies a very different problem:
>
> - The key research questions raised in this paper do not exist in federated learning, therefore the aggregator devised for federated learning  (Karimireddy et al., 2020) does not naturally address these questions. For example, in the paper our key research questions address the decentralized setting, and are: 1) Can regular workers reach consensus? 2) Can regular workers find minimizers/stationary point of their joint objective?  3) How does the communication topology pose the challenge to Byzantine robustness? These questions do not exist in federated learning where a server is available. Therefore, the clipping aggregator developed for federated learning  (Karimireddy et al., 2020) does not take topology into account and does not imply that it can address problems in decentralized learning.
>
> - The proof techniques in this paper are highly non-trivial. For instance, a corner case of our presented theoretic result by taking $\delta=0$ directly gives a convergence rate better than the state-of-the-art decentralized optimization (Koloskova et al., 2020b), even in absence of Byzantine workers.
>
> Compared to previous works in Byzantine-robust decentralized optimization, our work not only give better convergence rate but also applies to more topologies. For example, in Figure 8, we give an example of topology which does not satisfy the assumptions in previous works (Sundaram & Gharesifard, 2018; Su & Vaidya, 2016a) but satisfies our assumption. We defer the discussions to Appendix C.1 and Appendix F.
>
> We hope that this reply can address the novelty issue. If so, we would greatly appreciate it if you could raise the score accordingly to reflect above points.
>
>
> ### References
>
> Abadi, Martin, et al. "Deep learning with differential privacy." Proceedings of the 2016 ACM SIGSAC conference on computer and communications security. 2016.
>
> Chen, Xiangyi, Steven Z. Wu, and Mingyi Hong. "Understanding gradient clipping in private SGD: A geometric perspective." Advances in Neural Information Processing Systems 33 (2020): 13773-13782.
>
> Zhang, Jingzhao, et al. "Why gradient clipping accelerates training: A theoretical justification for adaptivity." arXiv preprint arXiv:1905.11881 (2019).

---

### Official Review · Reviewer_3Jpb · 2022-10-24

**Confidence:** 4
**Correctness:** 3
**Technical Novelty And Significance:** 2
**Empirical Novelty And Significance:** 3
**Recommendation:** 3

**Clarity, Quality, Novelty And Reproducibility:**

Regarding theory/statements/proofs:

-> Lemma 3. It should be $\forall i,j \in [n-b]$

-> Please justify Lemma 4.

-> In subsection E.1, bullets 4 and 5 have the same descriptions.

-> Page 27, Lemma 8 proof.  Applying inequality "(16)" to the last term, not inequality (18).

-> Page 29, Lemma 10 proof, I do not follow the second inequality when bounding $A_1$. Where does that square come from, and how does $\tau_i^{t+1}$ get into the argument?

-> Page 30, proof to theorem I': I do not understand why it is reasonable to apply Lemma 10 with $\eta = 0$. When the step size is 0, no update happens. As this bound needs to hold for every $t$, this step does not seem reasonable.

Regarding experiments:

-> It is not obvious that how the spectral gap $\gamma$ can be explicitly fixed or controlled. Could authors explain this issue?

-> In terms of comparison methods, I wonder if authors have thought of gradient tracking, which is a potential practice for decentralized optimization (ref. [A general framework for decentralized optimization with first-order methods by Xin et al., Proceedings of the IEEE (Volume: 108, Issue: 11, November 2020)]).

-> It is not immediately obvious to me how aggregators are implemented in this case. In the centralized setting, aggregators are usually applied on the server. I wonder if authors apply the aggregator on every node when they iterate?

**Strength And Weaknesses:**

Strength:

-> The paper is well-structured and the narrative flows finely.

Weakness:

-> There can be further clarification about the experimental settings.

-> Some proof steps do not naturally connect. Specific comments are in the following section.

-> I suggest the authors submit core code for numerical experiments.

**Summary Of The Paper:**

This submission discusses using the technique ClippedGossip in the decentralized optimization setting to counter potential Byzantine agents in the network.

The authors provide convergence guarantee in terms of first-order condition to show the attainment of consensus in the network, and give numerical experiments to show the effectiveness of the ClippedGossip technique in providing robustness in several attack/data heterogeneity settings.

**Summary Of The Review:**

There are several places where further clarification about theoretical proof and numerical experiments can be helpful for me to determine the correctness of the submission. It is not immediately clear to me that the selected numerical baseline is the most appropriate baseline, and I would like to examine the proof closer.

---

> ### Author Response · Authors · 2022-11-17
> **Author's response to Reviewer 3Jpb [Part 1/2]**
>
> We thank Reviewer 3Jpb for reading through the paper and especially through the proof and also for the suggestions. We address the concerns as follows and sincerely hope that in case our answers address your concerns you would adjust the score accordingly.
>
> ### Regarding weakness
>
> > There can be further clarification about the experimental settings.
>
> Since the experimental settings require detailed explanations, we have provided many of those in the appendix --- we describe the implementations in Appendix A, B, C and then provide the per-experiment settings in Appendix D.  More specifically,
>
> - In Appendix A,  we describe the details of aggregator implementations (e.g., CM, TM, RFA).
> - In Appendix B, we describe the details of attack implementations (e.g. ALIE, IPM, Dissensus).
> - In Appendix C.1, we describe the topologies (e.g. dumbbell, small-world, torus) used in the experiments.
> - In Appendix C.2, we describe the strategies to construct a mixing matrix in the presence of Byzantine workers.
> - In Appendix D, for full reproducibility, we first give the runtime hardware and software in Table 2 and default setup for MNIST experiment in Table 3 (e.g. neural net structure,  loss function, batch size, etc.). Then we give details (e.g. hyperparameters) for the experiments
>   - In Appendix D.1, we provide the details for Fig 3 in the main text.
>   - In Appendix D.2.1, we provide the details for Fig 4 in the main text.
>   - In Appendix D.2.2, we provide the details for Fig 6 in the main text.
>   - In Appendix D.2.4, we provide the details for Fig 5 in the main text.
>
> We provide additional experiments as well as their setup:
> - In Appendix D.2.3, we provide an experiment without “honest majority assumption”.
> - In Appendix D.3, we provide experiments on the CIFAR-10 dataset.
> - In Appendix D.4, we show that our algorithm relies on weaker topologies assumption than prior works.
> - In Appendix D.5, we give an experiment on the effect of constant clipping radius and adaptive clipping radius.
>
> We hope these settings are clear enough to the readers, together with the code provided (see below). But please do let us know if you have any other questions regarding the settings.
>
> > I suggest the authors submit core code for numerical experiments.
>
> We kindly note that on page 7 footnote 3, we have already provided a link to an anonymized repository where our code is shared.
>
> ### Regarding theory/statements/proofs:
>
> > Lemma 3. It should be ∀i,j∈[n−b]
>
> We thank the reviewer for pointing out the typo. We have fixed it in the revision.
>
> > Please justify Lemma 4.
>
> The Lemma 4 simply defines a spectral gap $\gamma$ for the matrix $\tilde{W}$. The spectral gap is $\gamma$ greater than 0 for connected graph (A1) and doubly stochastic mixing matrix $\tilde{W}$ in Lemma 3. The spectral gap is a common quantity to measure how well a topology diffuses information. For example, a fully connected graph with equal weight has the best spectral gap 1.
>
> Lemma 4 is common in the literature, such as for consensus (Boyd et al., 2006) and decentralized optimization (Koloskova et al., 2020).
>
> > In subsection E.1, bullets 4 and 5 have the same descriptions.
>
> We thank the reviewer for the proofreading. We have updated the descriptions to highlight the differences.
>
> > Page 27, Lemma 8 proof. Applying inequality "(16)" to the last term, not inequality (18).
>
> We thank the reviewer for pointing out the typo.
>
> > Page 29, Lemma 10 proof, I do not follow the second inequality when bounding A1. Where does that square come from, and how does tau+1 get into the argument?
>
> We have added the explanation for the second inequality in the proof (highlighted in blue).
>
> > Page 30, proof to theorem I': I do not understand why it is reasonable to apply Lemma 10 with η=0. When the step size is 0, no update happens. As this bound needs to hold for every t, this step does not seem reasonable.
>
> We kindly note that Theorem I’ considers the consensus problem, not the optimization problem. In the consensus problem, there is no gradient computation step, only the communication step. Therefore, a consensus step can be viewed as an optimization step with step size 0. We simplified the proof of Theorem I’ by reusing the analysis in Lemma 9 and 10 with $\eta=0$.

---

> > ### Author Response · Authors · 2022-11-17
> > **Author's response to Reviewer 3Jpb [Part 2/2]**
> >
> > #### Regarding experiments:
> >
> > > It is not obvious that how the spectral gap γ can be explicitly fixed or controlled. Could authors explain this issue?
> >
> > In Appendix C.2, we show how to construct the mixing matrix used by regular workers in general. For a given mixing matrix, one can explicitly compute the spectral gap $\gamma$ using linear algebra libraries or directly using its [definition (Wikipedia)](https://en.wikipedia.org/wiki/Spectral_gap) “the difference between the moduli of the two largest eigenvalues of a matrix”.
> >
> > For the experiment in Fig. 3, we provide the details in Appendix D.1. In this experiment, we consider the topology in Fig. 9 and manually design the mixing matrix and tune the weights to get the desired spectral gap.
> >
> > > In terms of comparison methods, I wonder if authors have thought of gradient tracking, which is a potential practice for decentralized optimization (ref. [A general framework for decentralized optimization with first-order methods by Xin et al., Proceedings of the IEEE (Volume: 108, Issue: 11, November 2020)]).
> >
> > While gradient tracking for decentralized optimization with non-iid data is very interesting, this paper is not directly comparable with our paper because we have quite different optimization objectives. The work of Xin et al. (2020) considers Byzantine-free decentralized training with undirected/directed graph and non-IID data and their goal is to find a minimizer/stationary point of the sum of all workers’ local objectives. In contrast, our Byzantine-robust decentralized optimization aims to find a minimizer/stationary point of the sum of (unknown) regular workers’ objectives. It is not trivial to us if (Xin et al., 2020) can be made Byzantine-robust and still enjoy a comparable convergence rate. It would be interesting for future work to consider Byzantine robustness with gradient tracking and also for directed graphs.
> >
> > > It is not immediately obvious to me how aggregators are implemented in this case. In the centralized setting, aggregators are usually applied on the server. I wonder if authors apply the aggregator on every node when they iterate?
> >
> > Yes, the aggregators are applied to every node. This is consistent with Byzantine-free decentralized optimization where the aggregator is gossip averaging and it is applied to every node. In our setting where Byzantine workers exist, we replace non-robust aggregators (gossip) with robust aggregators on every node.
> >
> >
> > We hope that Reviewer 3Jpb agrees to raise the score accordingly if our answers address the concerns raised regarding the proofs. We would be happy to answer any other questions if you have any.

---

### Official Review · Reviewer_vfZ6 · 2022-10-25

**Confidence:** 3
**Correctness:** 3
**Technical Novelty And Significance:** 2
**Empirical Novelty And Significance:** 3
**Recommendation:** 5

**Clarity, Quality, Novelty And Reproducibility:**

The work idea is original.  In general, the paper is easy to follow, except for the \tau issue.

**Strength And Weaknesses:**

+ The algorithm dose not rely on additional graph connectivity
+ The performance is better than some existing algorithms

- The algorithm does not guarantee accuracy of the optimization process
- The idea behind the algorithm is unclear. In particular, the Clipping idea involves a parameter \tau but its role is not well explained and illustrated.
- The paper has an adaptive selection process for \tau_i, but it cannot be computed only rely on local information. In particular, it involves \delta_max, which seems not an available information.

**Summary Of The Paper:**

The paper proposes a novel algorithm to deal with Byzantine resilient decentralized optimization for data training.

**Summary Of The Review:**

The paper proposes a novel algorithm to deal with Byzantine resilient decentralized optimization for data training, which has better performance compared with some existing works. However, the idea behind the algorithm is not clearly explained. In particular, the role of the \tau parameter and the adaptive selection process for \tau_i is unclear to me.

---

> ### Author Response · Authors · 2022-11-17
> **Author's response to Reviewer vfZ6**
>
> We thank Reviewer vfZ6 for the useful feedback. We address the concerns as follows. We sincerely hope that in case our answers address your concerns you would adjust the score accordingly.
>
> > The algorithm does not guarantee accuracy of the optimization process
>
> For Byzantine-robust optimization with non-IID data (assumption A3), there is a lower bound for **ALL** algorithms because byzantine workers can be indistinguishable from heterogeneous data (Karimireddy et al., 2021). It means no algorithm can “guarantee accuracy” and therefore this is not a disadvantage of our algorithm. On the other hand, our algorithm achieves superior convergence rates compared to other robust decentralized learning algorithms, even faster than Byzantine-free decentralized learning, c.f. Table 1.
>
> If Assumption A3 is strengthened to the “strong growth condition” where the model is overparameterized, then regular workers with heterogeneous data share the same minimizer, similar to the IID case. Under this additional assumption of (Karimireddy et al., 2021b), our lower bound could be also made 0 and the accuracy can be guaranteed. The proof is deferred as future work.
>
> > The idea behind the algorithm is unclear. In particular, the Clipping idea involves a parameter \tau but its role is not well explained and illustrated.
>
> We thank the reviewer for bringing up the explanation of the clipping idea. **We added a diagram for ClippedGossip in Fig. 17 and explain the idea of the algorithm**. More concretely, in the context of Byzantine-resilient optimization, a Byzantine worker can send very large gradients/updates to compromise a model. In our algorithms, the gradients/updates are clipped before aggregation so that they have limited influence on the model. A smaller clipping radius can ensure robustness but a too-small one will slow down convergence. So ideally, we would like to use a clipping radius that is neither too large nor too small. To find a practical clipping radius, we propose an adaptive strategy in Eq. (10) on p.24 which only uses information available to the worker.
>
> In Section D.5 Fig16, we compare different constant clipping radiuses and the adaptive clipping radius strategy. The experiment results suggest that the adaptive strategy work has overall good performance across all attacks.
>
> > The paper has an adaptive selection process for \tau_i, but it cannot be computed only rely on local information. In particular, it involves \delta_max, which seems not an available information.
>
> We can know an upper bound of $\delta_\max$ through either prior knowledge or computation. In federated learning, while the exact number of Byzantine workers is unknown, the number of Byzantine workers is upper bounded based on assumptions or prior knowledge. For example, an honest majority assumption as used widely in the literature guarantees that $\delta<0.5$; in some applications, there is prior knowledge or empirical evidence serving as an upper bound on $\delta$. On the other hand, in decentralized learning with more than 1 regular worker and Assumption (A1), $\delta_i$ can be strictly upper bounded by $1-W_{ii}$ and so $\delta_\max$ can be strictly bounded by $1-\max_i{W_{ii}}$. These upper bounds can be explicitly known to all regular workers, using the constructions described in Appendix C.2. So even if we don’t have prior knowledge, we can use an upper bound of $\delta_\max$ when choosing the clipping radius.
>
> On p.24 Eq. (10) and the equation before Eq. (10), we give an empirical and adaptive selection strategy for $\tau$ which only uses the local information. We note that the theoretical clipping radius given in Eq. (27) is not the only possible sufficient radius --- there are a range of clipping radiuses that attain the same convergence rate. This is because the $\tau$ in Eq. (27) is given by Lemma 10 which minimizes an upper bound for $\Xi$. This upper bound is not tight and therefore admits a range of possible $\tau$. The $\tau$ that minimizes $\Xi$ would be smaller than Eq. (27). Therefore, we use Eq. (10) which gives a bit smaller $\tau$ than Eq. (27) and only uses locally available information. More specifically, we use the upper bound of $\delta_\max$ and pick a quantile of the smallest distances to determine the clipping radius, without needing to know the identities of regular neighbors.
>
> The empirical performance in Fig. 16 shows that the given adaptive strategy has good performance against different attacks.
>
> We sincerely hope the reviewer could raise the score if our answer addresses your concerns.

---

### Decision · Program_Chairs · 2023-01-20

**Decision:**

Reject

**Justification For Why Not Higher Score:**

NA

**Justification For Why Not Lower Score:**

NA

**Metareview: Summary, Strengths And Weaknesses:**

This work studies clipping of gossip style algorithms for decentralized optimization problems in order to mitigate the effect of byzantine agents. Convergence analysis is provided and experiments corroborate the merits of the proposed approach.

Strengths are in establishing robustness to byzantine agents.

Weaknesses are in failing to rigorously contrast with prior approaches to establish robustness to byzantine agents, and that many of the technical results presented here are already known or exceptionally similar.

**Summary Of Ac-Reviewer Meeting:**

NA